# Emission of nitrous acid from soil and biological soil crusts represents an important source of HONO in the remote atmosphere in Cyprus

Hannah Meusel[1], Alexandra Tamm[1], Uwe Kuhn[1], Dianming Wu[1,#], Anna Lena Leifke[1], Sabine Fiedler[2], Nina Ruckteschler[1], Petya Yordanova[1], Naama Lang-Yona[1], Mira Pöhlker[1], Jos Lelieveld[3,4], Thorsten Hoffmann[5], Ulrich Pöschl[1], Hang Su[6,1], Bettina Weber[1], Yafang Cheng[1,6]

[1]Max Planck Institute for Chemistry, Multiphase Chemistry Department, Mainz, Germany
[2]Johannes Gutenberg University, Institute for Geography, Mainz, Germany
[3]Max Planck Institute for Chemistry, Atmospheric Chemistry Department, Mainz, Germany
[4]The Cyprus Institute, Nicosia, Cyprus
[5]Johannes Gutenberg University, Institute for Inorganic and Analytical Chemistry, Mainz, Germany
[6]Institute for Environmental and Climate Research, Jinan University, Guangzhou, China
[#]now at: School of Geographic Sciences, East China Normal University, Shanghai, China

Corresponding author: Hang Su (h.su@mpic.de) and Bettina Weber (b.weber@mpic.de)

**Abstract.** Soil and biological soil crusts can emit nitrous acid (HONO) and nitric oxide (NO). The terrestrial ground surface in arid and semi-arid regions is anticipated to play an important role in the local atmospheric HONO budget, deemed to represent one of the unaccounted HONO sources frequently observed in field studies. In this study HONO and NO emissions from a representative variety of soil and biological soil crust samples from the Mediterranean island Cyprus were investigated under controlled laboratory conditions. A wide range of fluxes was observed, ranging from 0.6 to 264 ng m$^{-2}$ s$^{-1}$ HONO-N at optimal soil water content (20-30% of water holding capacity, WHC). Maximum NO-N fluxes at this WHC were lower (0.8-121 ng m$^{-2}$ s$^{-1}$). Highest emissions of both reactive nitrogen species were found from bare soil, followed by light and dark cyanobacteria-dominated biological soil crusts (biocrusts), correlating well with the sample nutrient levels (nitrite and nitrate). Extrapolations of lab-based HONO emission studies agree well with the unaccounted HONO source derived previously for the extensive CYPHEX field campaign, i.e., emissions from soil and biocrusts may essentially close the Cyprus HONO budget.

## 1 Introduction

Nitrous acid (HONO) plays an important role in tropospheric chemistry, as it is one of the major precursors of the hydroxyl (OH) radical which determines the oxidizing capacity of the atmosphere. In the early morning, HONO photolysis has been shown to contribute up to 30% to the local OH budget (Alicke et al., 2002; Kleffmann et al., 2005; Ren et al., 2003 and 2006; Meusel et al., 2016). Currently, the HONO formation processes, especially during daytime, are still not fully understood. Recent ground based field measurements showed unexpected high daytime concentrations of HONO, which could not be solely explained by atmospheric gas phase reactions (R1-R3) (Kleffmann et al., 2003 and 2005; Su et al., 2008a; Sörgel et al., 2011a; Su et al., 2011; Michoud et al., 2014; Czader et al., 2012; Wong et al., 2013; Tang et al., 2015; Oswald et al., 2015, Meusel et al., 2016).

$$OH + NO \rightarrow HONO \tag{R1}$$

$$HONO \xrightarrow{\text{hv (300-405 nm)}} OH + NO \qquad (R2)$$

$$HONO + OH \rightarrow NO_2 + H_2O \qquad (R3)$$

Several studies have shown that HONO can be heterogeneously formed from $NO_2$ on a variety of surfaces, e.g., soot, humic acid, minerals, proteins and organically coated particles (Ammann et al., 1998; Arens et al., 2001; Aubin et al., 2007; Bröske et al., 2003; Han et al., 2013; Kalberer et al., 1999; Kleffmann et al., 1999; Kleffmann and Wiesen, 2005; Lelievre et al., 2004; Kinugawa et al., 2011; Liu et al., 2015; Wang et al., 2003; Yabushita et al., 2009; Meusel et al., 2017). Light can activate some of these surfaces (humic acid, proteins and other organic compounds, titanium dioxide, soot), which enhances $NO_2$ uptake and HONO production (George et al., 2005; Langridge et al., 2009; Monge et al., 2010; Ndour et al., 2008; Ramazan et al., 2004; Stemmler et al., 2007; Kebede et al., 2013; Meusel et al., 2017). But $NO_2$ uptake coefficients and the ambient aerosol surface areas for heterogeneous reactions of $NO_2$ were nevertheless frequently found to be too low to account for the observed HONO production rates (Stemmler et al., 2007; Sarwar et al., 2008; Zhang et al., 2016). Besides the heterogeneous $NO_2$ reaction, Bejan et al. (2006) observed HONO formation during irradiation of nitrophenols. Photolysis of nitrate or nitric acid generates HONO as well (Baergen and Donaldson, 2013; Scharko et al., 2014; Zhou et al., 2003, 2011). Contrary to the detected missing HONO source near the ground, recent airborne measurements (500-1200 m above ground level) observed HONO concentrations, which could be explained by gas phase reactions only (Li et al., 2014; Neuman et al., 2016). However, vertical gradient studies showed higher HONO concentrations near the ground than in higher altitudes indicating a ground level source (Harrison and Kitto, 1994; Kleffmann et al., 2003; Ren et al., 2011; Stutz et al., 2002; VandenBoer et al., 2013; Villena et al., 2011; Zhou et al., 2011; Wong et al., 2012 and 2013; Vogel et al., 2003; Zhang et al., 2009; Young et al., 2012). This is supported by gas exchange studies showing that HONO and NO can be emitted from (natural) soil and biological soil crusts (biocrusts, BSC), even without applying atmospheric $NO_2$ (Su et al., 2011; Oswald et al., 2013; Mamtimin et al., 2016; Weber et al., 2015; Meixner and Yang, 2006). HONO and NO can be formed during biological processes (nitrification and denitrification; Pilegaard, 2013), in which $NH_3$ or $NH_4^+$ is oxidized stepwise or $NO_3^-$ is reduced (Fig. 1). Depending on soil-pH and according to Henry´s law soil nitrite ($NO_2^-$) can be converted into gaseous HONO. It was found that sterilized soil emit lower amounts of reactive nitrogen than natural soil (Oswald et al., 2013; Weber et al., 2015).

Biocrusts grow within the uppermost millimeters to centimeters of soil in arid and semi-arid ecosystems. They are composed of photoautotrophic cyanobacteria, algae, lichens, and bryophytes, growing together with heterotrophic bacteria, fungi and archaea in varying proportions (Belnap et al., 2016). Depending on the dominating photoautotrophs, cyanobacteria-dominated biocrusts with an initial thin light-colored and a well-developed dark type, cyanolichen- and chlorolichen-dominated biocrusts with lichens comprising cyanobacteria or green algae as photobionts, and bryophyte-dominated biocrusts are distinguished (Büdel et al., 2009). Many free living cyanobacteria but also those in symbiosis with fungi (forming lichens) and vascular plants can fix atmospheric nitrogen $N_2$ and convert it into ammonia (Cleveland et al., 1999; Belnap 2002; Herridge et al., 2008; Barger et al., 2016). Globally it has been estimated that 100-290 Tg (N) $yr^{-1}$ is fixed biologically (Cleveland et al., 1999), of which 49 Tg $yr^{-1}$ (17-49%) is fixed by cryptogamic covers, which comprise biocrusts, but also other microbially dominated biomes, like lichen and bryophyte communities occurring on soil, rocks and plants in boreal and tropical regions (Elbert et al., 2012). Studies have suggested, that nitrogen cycling in soil ($N_2$ fixation, nitrification, denitrification)

and hence reactive nitrogen emission (NO, $N_2O$, HONO) is often enhanced by well-established biocrusts, especially
by dark cyanobacteria (Cleveland et al., 1999; Elbert et al., 2012; Belnap, 2002; Barger et al., 2013; Johnson et al.,
2005; Abed et al., 2013; Strauss et al., 2012; Weber et al., 2015). But much of the molecular biology/chemistry that
is important for atmosphere-land interactions is likely occurring just below the crust (that is visible at the surface).
In Cyprus, an island in the semi-arid eastern Mediterranean area, biocrusts are ubiquitously covering ground surfaces
and hence can be anticipated to play an important role in the local HONO budget. In the CYPHEX campaign 2014
(CYprus PHotochemical EXperiment) the observed diel cycles of HONO ambient air concentrations revealed strong
unaccounted sources of HONO and NO, being well correlated with each other (Meusel et al., 2016). With low $NO_2$
concentrations and high $HONO/NO_x$ ratios, respectively, direct emissions from combustion and heterogeneous
reactions of $NO_2$ could be excluded as significant HONO sources, leaving emissions from soil and the respective
surface cover to be the most plausible common source for both nitrogen species (Meusel et al., 2016).
In the present study we have measured HONO and NO fluxes from soil and biocrust samples from Cyprus by means
of a dynamic chamber system. The aim of this study was to characterize and quantify direct trace gas emissions and
demonstrate their impact on the atmospheric chemistry in the remote coastal environment of Cyprus.
**2 Methods**
**2.1 Sampling**
Bare soil and biocrust samples were collected on 27th April 2016 on the South/South-East side of the military station
in Ineia, Cyprus (34.9638°N, 32.3778°E), where the CYPHEX campaign took place in 2014. It is a rural site about
600 m above sea level (asl), approximately 5-8 km from the coast and is surrounded by typical Mediterranean
vegetation (olive and pine trees, small shrubs like *Pistacia lentiscus, Sacopoterium spinosum* and *Inula viscosa).*
More details about the site can be found in Meusel et al. (2016).
In an area of about 8580 m² (South/South-East direction of the station) 50 grids (25x25 cm) were placed at randomly
selected spots for systematic ground cover assessment. At each grid point occurrence of nine types of surface cover
(i.e., light and dark cyanobacteria-, chlorolichen-, cyanolichen-, and moss-dominated biocrust, bare soil, stone, litter,
vascular vegetation/shrub) were assigned and quantified. Spatially independent replicate samples were collected of
light cyanobacteria-dominated biocrusts (light BSC), dark cyanobacteria-dominated biocrusts with cyanolichens
(dark BSC), chlorolichen-dominated biocrusts (chlorolichen BSC I, chlorolichen BSC II), moss-dominated biocrusts
(moss BSC) and of bare soil (Fig. S1 of the supplement). Each sample was collected in dry state in a plastic petri
dish (diameter 5.5. cm, height 1 cm), sealed and stored in the dark at room temperature until further analysis (storage
time less than 15 weeks). Storage of biocrust samples under dry and dark conditions at room temperature is the most
widely spread method, and has been used in many other studies on N-cycling in which samples have been stored
even up to 6 month before measurements were performed (Abed et al., 2013; Strauss et al., 2012:, Johnson et al.,
2007; Brankatschk et al., 2013).
In total 43 samples were collected (Table 1) of which 18 samples, i.e., 3 replicates of each HONO emitting surface
cover type were used directly (upfront) for nutrient analysis, while all others were first used for trace gas exchange
measurements, prior to nutrient and chlorophyll content analysis.
**2.2 Meteorological data**
During CYPHEX the meteorological parameters were measured at about 5 m above ground, considered not
representative for the micro-habitat of the soil ground surface. Hence we placed three humidity (and temperature)
sensors (HOBO Pro v2) just on top of the soil surface about 4 weeks prior to sample collection. Reference
meteorological data (air temperature, humidity and precipitation) from Paphos airport (about 20 km south of the
sample area, 12 m asl) and Prodromos (about 40 km east of the sampling area, 1380 m asl) during the sampling
period as well as the precipitation data from the last 4 years (2013-2016) were provided by the Department of
Meteorology, Cyprus
(http://www.moa.gov.cy/moa/ms/ms.nsf/DMLmeteo_reports_en/MLmeteo_reports_en?opendocument; last access:
Dec. 2016).
**2.3 Soil characteristics: nutrient, chlorophyll and pH**
Soil characteristics (nutrient, pH) have an effect on soil emission, e.g., higher nutrient level and lower pH would
enhance emission according to Henry law (Su et al., 2011). Nutrient analysis was conducted on samples without gas
exchange measurements (n = 3) and on replicate samples after gas exchange measurements in order to analyze
potential effects of the applied 'wetting-drying' cycle. Nitrate ($NO_3^-$), nitrite ($NO_2^-$) and ammonium ($NH_4^+$) were
analyzed via flow injection analysis with photometric detection (FIAstar 5000, Foss, Denmark). Prior to that, the
samples comprised of soil and its biocrust-cover were gently ground and an aliquot of 7 g was solved in 28 mL of
0.0125 M $CaCl_2$. After shaking for 1 hour the mixture was filtered on a N-free filter. The detection limits were 0.014,
0.046 and 0.047 mg kg$^{-1}$ for $NO_2^-$-N, $NO_3^-$-N and $NH_4^+$-N, respectively.
Chlorophyll analysis, as an indicator of biomass of photo-autotrophic organisms, was done according to the dimethyl
sulfoxide (DMSO) method (Ronen and Galun, 1984). Ground samples were extracted twice with $CaCO_3$ saturated
DMSO (20 mL, 10 mL) at 65°C for 90 min. Both extracts were combined and centrifuged (3000 G) at 15°C for 10
min. The light absorption at 648, 665 and 700 nm was detected with a spectral photometer (Lambda 25 UV/VIS
Spectrometer, Perkin Elmer, Rodgau). The amount of chlorophyll a ($Chl_a$) was calculated according to Arnon et al.
(1974). Chlorophyll a+b ($Chl_{a+b}$) content was calculated according to Lange, Bilger and Pfanz (pers. comm. in Weber
et al., 2013):
$$Chl_{a+b}[\mu g] = \left(20.2 \cdot (E_{648} - E_{700}) + 8.02 \cdot (E_{665} - E_{700})\right) \cdot a \qquad \text{(eq.1)}$$

$$Chl_a[\mu g] = \left(12.19 \cdot (E_{665} - E_{700})\right) \cdot a \qquad \text{(eq.2)}$$

where $Chl_{a+b}[\mu g]$, $Chl_a[\mu g]$ is the chlorophyll content of the sample, $E_{648}$, $E_{665}$, $E_{700}$ are light absorption at the given
wavelength, and a is the amount of DMSO used in mL.
The pH was determined for each surface cover type (n = 3-4) according to Weber et al. (2015, Suppl.). Here, 1.5 g of
the ground sample was mixed with 3.75 mL of pure water and shaken for 15 min. Then the slurry was centrifuged
(3000 G, 5 min) to separate the solid phase from the liquid solution. The latter was used for pH determination by
means of a pH electrode (Inlab Export Pro-ISM, Mettler Toledo).

**2.4 Trace gas exchange measurements**

The dynamic chamber method for analyzing NO and HONO emissions from soil samples was already introduced before (Oswald et al., 2013; Weber et al., 2015; Wu et al., 2014) and in general showed good agreement with flux measurements in the field (van Dijk et al., 2002; Rummel et al., 2002). Under the prevailing dry and hot conditions in Cyprus macropores and cracks are likely to be present in the soil layer. It is assumed that during the sampling and sample treatment the number and sizes of soil cracks was not significantly increased so that the diffusivity of gases in the soil samples is comparable to the one in soil in the natural environment. Intact soil and biocrust samples (25-35 g in a plastic petri dish with 5.5 cm diameter and about 1 cm height) were wetted with 8-13 g of pure water (18.2 M$\Omega$) up to full water holding capacity and placed into a dynamic Teflon film chamber ($\approx$47 L) flushed with 8 L min$^{-1}$ dry pure air (PAG 03, Ecophysics, Switzerland). Intact (biocrust) samples consist of a few mm of the biocrust and the underlying soil. Typical drying cycles lasted between 6 and 8 hours. A Teflon coated internal fan ensured complete mixing of the chamber headspace volume. During the experiments the chamber was kept at constant temperature (25°C, the mean daytime air temperature during CYPHEX) and in darkness to avoid photochemical reactions. At the chamber outlet the emitted gases HONO, NO and water vapor were quantified. HONO was analyzed with a commercial long path absorption photometer (LOPAP, QUMA GmbH; Wuppertal, Germany), with a detection limit of ~4 ppt and 10% uncertainty (based on the uncertainties of liquid and gas flow, concentration of calibration standard and regression of calibration). To avoid any transformation of HONO in the tubing, the sampling unit including the stripping coil from LOPAP was directly connected to the chamber. $NO_x$ (NO + $NO_2$) was detected with a commercial chemiluminescence detector (42i TL, Thermo Scientific; Watham, USA) modified with a photolytic converter with a detection limit of ~50 ppt (NO) and ~200 ppt ($NO_2$). An infrared $CO_2$ and $H_2O$ analyzer (Li-840A, LICOR; Lincoln, USA) was used to log the drying and to calculate the soil water content (SWC) of the samples as follows:

$$SWC(WHC) = \frac{m_{H2O,t=n}}{m_{H2O,0}} * 100 \qquad \text{(eq. 3)}$$

$$m_{H2O,t=n} = m_{H2O,t=n-1} - \frac{S_{Licor,t=n}}{\sum_{t=0}^{t=N} S_{Licor}} * m_{H2O,0} \qquad \text{(eq. 4)}$$

with t=0 denoting the measurement start (wetted sample inserted into chamber), t=n: any time between 0 and N, t=N: time when sample had dried out and measurement was stopped, $S_{Licor}$: absolute $H_2O$ signal at a given time, $m_{H2O,0}$: mass of water added to sample (water holding capacity, WHC), SWC: soil water content in % WHC.

**2.5 Data analysis**

Measured data of $NO_2^-$, $NO_3^-$, $NH_4^+$, $Chl_{a+b}$, $Chl_a$, NO and HONO optimum flux and NO and HONO integrated flux did not follow a normal distribution. Rather, log-transformed data were normally distributed (Shapiro-Wilk) and therefore used for statistical analysis (Pearson correlation, ANOVA including Tukey Test with significance level of p = 0.05) executed with OriginPro (version 9.0; OriginLab coporation, Northampton, Massachusetts, USA). Precipitation data from the last 4 years (2013-2016), provided by the Department of Meteorology of Cyprus, indicating about 30 rain events per year (precipitation > 1 mm with following one or more dry days) were used to estimate annual emissions of total nitrogen in terms of HONO and NO.

**3 Results and discussion**
**3.1 Meteorological conditions**
One month before sampling, three sensors measuring temperature and relative humidity were installed directly above
the soil surface in the field to represent the micro-climate of the ground surface. Reference air temperature, humidity
and precipitation measurements at Paphos airport and Prodromos showed one rain event on 11-12 April which is
reflected by higher soil humidity (80-100%) and lower temperatures on these days (see Fig. S2). As a consequence,
the biological soil crusts were activated and went through one full wetting and drying cycle before sample collection.
Temperature above the soil ranged from 10°C in the night to 50°C during the day when solar radiation was most
intense. Air temperature was similar during the night but not as hot during the day ranging between 20° and 30°C.
Humidity above the ground was low during daytime (<30% rH) and increased during the night up to 80%, while the
atmospheric relative humidity (at Paphos airport) ranged between 47 and 73% (without rain event). Thus there were
only little variations of humidity with height above the soil surface. Above the ground surface the relative humidity
was somewhat lower during the day (mainly caused by higher temperatures) but somewhat higher during the night,
compared to respective weather station data. During and shortly after the main rain event humidity at ground level
was higher (80 and 100% rH) compared to ambient air humidity (70-85% rH). Ambient air temperatures were
somewhat lower during sample collection of this study as compared to the CYPHEX field campaign in 2014. During
CYPHEX, nighttime temperatures (3 m above ground level) did not drop below 18°C. Relative humidity (3 m above
ground level) was mostly between 70 and 100% with only two short periods with humidity between 20-60% rH.
Hence we can assume that soil surface temperatures were higher and ground rH in the same range during CYPHEX
compared to sampling period.
**3.2 Cyprus soil and biocrust characteristics**
The different biocrust types were distinguished in the field based on the dominating phototrophic compound but
which provides no information about the microbial community below or about the magnitude of (de)nitrification
processes. The microbial community couldn´t be determined by non-destructive methods. Systematic mapping of
surface covers revealed that moss-dominated biocrusts are the most frequent in the investigated Cyprus field site area
(21.3%), followed by light (10.4%) and dark BSC (6.5%), whereas chlorolichen- (3.2%) and cyanolichen-dominated
BSC (1.8%) only played a minor role (Fig. 2, Fig. S1). The soil surface was partially covered by litter (26.3%),
stones (19.5%) and vascular vegetation (8.5%), whereas open soil was rarely found (2.5%). It was previously
established that soil and biocrusts emit HONO and NO (Weber et al., 2015; Oswald et al., 2013), jointly accounting
for 45.6% of surface area in our studied region. To the best of our knowledge, no data on reactive nitrogen emissions
from vascular vegetation and plant litter have been published yet.
Nutrient analysis revealed large variations in concentrations of nitrogen species ranging from 0 to 6.48, 0 to 0.57 and
0 to 22.2 mg (N) kg$^{-1}$ of dry soil/crust mass for $NO_3^-$, $NO_2^-$, and $NH_4^+$, respectively (Fig. 3a, Tab. S1 of the
supplement). In general, no significant change in reactive nitrogen contents was found before and after the trace gas
exchange experiments, indicating no significant impact of one wetting-drying cycle on the nutrient content. Bare soil
samples had significantly higher levels of $NO_3^-$ and $NO_2^-$ content compared to dark, chlorolichen and moss BSC.

Among the latter three, no significant differences in nutrient levels were observed. Light BSC had $NO_2^-$ contents similar to bare soil. The $NH_4^+$ content was very similar in all samples, except for one outlier in the group of light BSC with strongly elevated $NH_4^+$. Higher nitrate and ammonium levels in bare soil compared to crust-covered samples were also reported recently for a warm desert site in South Africa (Weber et al., 2015), indicative of nutrient consumption/integration by the biocrusts. Nitrite, on the other hand, was lower for bare soil samples compared to biocrust samples. While $NO_3^-$ was slightly higher, $NH_4^+$ and $NO_2^-$ contents (especially of bare soil samples) were lower in the South African arid ecosystem compared to Cyprus.

Chlorophyll was only determined in the samples used for flux measurements. $Chl_a$ ranged from 4.1 (bare soil) to 144.2 mg m$^{-2}$ (moss BSC) and $Chl_{a+b}$ from 9.3 (bare soil) to 211.3 mg m$^{-2}$ (moss BSC), respectively (Fig. 3b, Tab. S1). From bare soil, via light BSC and chlorolichen BSC II, to dark BSC the chlorophyll content increased, but not significantly (p > 0.2). Nevertheless, $Chl_a$ and $Chl_{a+b}$ contents of chlorolichen BSC I and moss BSC were significantly higher than these of bare soil, light BSC and chlorolichen BSC II (p<0.05, Fig. 43).. The range of chlorophyll contents is comparable to previous arid ecosystem studies (Weber et al., 2015).

The pH of soil and biocrusts ranged between slightly acidic (6.2) and slightly alkaline (7.6; Fig. 3c). The mean pH of 17 samples was 7.0, i.e., neutral. Only the pH of moss BSC samples was significantly lower than that of bare soil, light BSC and chlorolichen BSC samples (p=0.05). Soil and biocrust samples from South Africa were slightly more alkaline (7.1-8.2) with no significant difference among biocrust types (Weber et al., 2015).

**3.3 NO and HONO flux measurements**

All samples showed HONO and NO emissions during full wetting and drying cycles. The calculations of the emission or flux rates are shown in the supplement. Maximum emission rates of HONO were observed at about 17-33% WHC, and of NO at 20-36% with no significant differences between all soil cover types (Fig 4). Emissions declined to zero at 0% WHC and to very small rates for >70%. Emission maxima strongly varied between soil cover types, but also between samples of the same cover type (see Fig. 4 and 5, and Table S1). Highest emissions of both HONO-N and NO-N were detected for bare soil (175 ± 50.4 and 92.2 ± 20.0 ng m$^{-2}$ s$^{-1}$; values indicate mean ± standard error), followed by light (48.6 ± 24.3 and 44.0 ± 22.4 ng m$^{-2}$ s$^{-1}$) and dark BSC (27.1 ± 16.1 and 26.5 ± 15.9 ng m$^{-2}$ s$^{-1}$). Both types of chlorolichen- and moss-dominated biocrusts showed very low emission rates of reactive nitrogen (on average < 10 ng m$^{-2}$ s$^{-1}$). Maximum HONO emissions were somewhat higher than maximum NO emissions, especially for bare soil. Integrating full wetting and drying cycles (6-8 hours), 0.04-1.9 mg m$^{-2}$ HONO-N and 0.06-1.6 mg m$^{-2}$ NO-N were released (Fig. 5, lower panel). While the maximum fluxes of reactive nitrogen emission were higher for HONO than NO, especially from bare soil, the integrated emissions were similar or even larger for NO, which is released over a wider range of SWC.

In general, it is difficult to compare chamber flux measurements of different studies due to different experimental configurations, such as chamber dimension, flow rate, resident time and drying rate etc. Here, we compared our results to studies which applied the same method (with the same or very similar conditions). The emission rates were consistent with these studies where HONO-N or NO-N emissions from soil between 1-3000 ng m$^{-2}$ s$^{-1}$ were found (Su et al., 2011; Oswald et al., 2013; Mamtimin et al., 2016; Wu et al., 2014; Weber et al., 2015). Mamtimin et al. (2016) observed NO-N fluxes at 25°C of 57.5 ng m$^{-2}$ s$^{-1}$, 18.9 ng m$^{-2}$ s$^{-1}$ and 4.1 ng m$^{-2}$ s$^{-1}$ for soil of grape and cotton

fields and desert soil from an oasis in China, respectively. Oswald et al. (2013) found HONO-N and NO-N emissions between 2 and 280 ng $m^{-2}$ $s^{-1}$ (each) from different soil from all over the world covering a wide range of pH, nutrient content and organic matter. Biogenic NO emissions of 44 soil samples from arid and semi-arid regions were reviewed by Meixner and Yang (2006) with N-fluxes ranging from 0 to 142 ng $m^{-2}$ $s^{-1}$.

In contrast to the results of the present study, where bare soil showed highest emissions, Weber et al. (2015) found lowest emission from bare soil in samples from South Africa. In that study, dark cyanobacteria-dominated biocrusts revealed highest emission rates (each HONO-N and NO-N up to 200 ng $m^{-2}$ $s^{-1}$), followed by light cyanobacteria-dominated biocrusts (up to 120 ng $m^{-2}$ $s^{-1}$), whereas in the present study, emissions of dark cyanobacteria-dominated biocrusts tended to be lower. No significant difference of HONO-N and NO-N emissions from light BSC between both sample origins were found. HONO-N and NO-N emissions of moss- and chlorolichen-dominated biocrusts were low in both studies (each <60 ng $m^{-2}$ $s^{-1}$) but still significantly higher for samples from South Africa than from Cyprus. In the present study HONO maximum emissions were higher than for NO (while integrated emissions being comparable) while in the study of Weber et al. (2015) HONO maximum fluxes were somewhat lower than those of NO. The present results of nitrogen emissions correlated well with the nutrient contents (especially $NO_2^-$ and $NO_3^-$, Fig. 6). Bare soil, in which highest $NO_3^-$ and $NO_2^-$ levels were found, also showed highest HONO and NO emissions. A good linear correlation was found between $NO_2^-$ contents and emission of both nitrogen gas phase species for all samples ($R^2$ = 0.84 for HONO and 0.85 for NO; p<0.001). The level of correlation between $NO_3^-$ and HONO and NO was lower, but still significant ($R^2$ = 0.68 and 0.67, respectively, p<0.001). Low correlations were found between HONO or NO emissions and $NH_4^+$-contents ($R^2$ = 0.165 and 0.232; p=0.05). Thus, in the present study it seems that reactive nitrogen emissions predominantly depend on $NO_2^-$ and $NO_3^-$ contents and not on surface cover types, although biocrusts (especially with cyanobacteria and cyanolichens) are able to fix atmospheric nitrogen (Belnap, 2002; Elbert et al., 2012; Barger et al., 2013; Patova et al., 2016). The results of a two-factorial ANOVA showed that HONO or NO emissions were not significantly related to soil cover type but rather with nitrite content, i.e., its direct aqueous precursor. For nitrate, the two-factorial ANOVA indicated dependencies of both cover type and nutrient content. Long range transport and instantaneous atmospheric deposition of NOx and nitrate/nitrite/ammonium can be excluded to be a dominant source of HONO and NO precursors in local soil, as the observed concentrations in Cyprus ambient air were very low (Meusel et al., 2016; Kleanthous et al., 2014). A dominant contribution from microbial activity to the nutrient content is anticipated, although long-term atmospheric accumulation of nutrients in the soil prior to the field campaign cannot be excluded. While biocrusts increase nutrient availability via N fixation, it is their possible associations with ammonia oxidizing microbes (bacterial and archaea) that finally convert the fixed nitrogen to nitrite and nitrate. Determination of the microbial community below the biocrust or in bare soil was not carried out as it was outside the scope of this study. Nitrification and other nitrogen cycling processes are not restricted to biocrusts, but can also occur in non-crusted soils. The relevance of these processes is expected to depend on substrate richness (i.e. amount of ammonium available for nitrifiers). Our results differ from those obtained by Weber et al. (2015) on South African samples, as there HONO and NO emissions were not correlated with bulk concentrations of ammonium, nitrite and nitrate. In their study nitrite content was lowest for bare soil compared to other biocrust types. Ammonium and nitrite levels were also lower than in the present study. Therefore Weber et al. (2015) indicated that biocrusts can enhance N-cycle and emission of reactive nitrogen.

Since most of the samples were slightly alkaline and only moss samples were slightly acidic, no effect of pH could
be observed. But in general it is expected that with higher nutrient and lower pH values HONO emission is increased
by simple partitioning processes (Su et al., 2011). The simulated equilibrium concentration at soil surface [HONO]*
(equation see Su et al., 2011) is much lower than the measured one. This deviation is probably based on the non-ideal
behavior of the soil samples (adsorption, Kelvin and solute interaction effects on gas/liquid partitioning). But this
method does not allow argumentation on physical or biological processes.
**3.4 Comparison of soil emission and observed missing source**
To quantify the flux rate of HONO emissions from soil to the local atmosphere and to compare it to the unaccounted
source found in Cyprus in 2014 (Meusel et al., 2016), we applied a standard formalism describing the atmosphere-
soil exchange of trace gases as a function of the difference between the atmospheric concentration and the
equilibrium concentration at the soil solution surface [HONO]* (Su et al., 2011):
$$F^* = v_T \left( [HONO]^* - [HONO] \right) \quad\quad\quad (eq.5)$$
where [HONO] is the ambient HONO concentration measured on Cyprus (mean daytime average 60 ppt) and
[HONO]* is the equilibrium concentration at soil surface. [HONO]* can be determined from measurements in a
static chamber. In a dynamic chamber system, there is a concentration gradient of HONO between the headspace
(where HONO was measured) and the soil surface. Here we used the measurements of water vapor to correct for the
soil surface concentration and equilibrium concentration of HONO by assuming a similar gradient for the two
species. A correction coefficient of 3.8 was determined, which is the ratio of the equilibrium rH of 100% over wet
soil surface to the initial headspace rH of 25-30% after inserting the wet sample into the chamber. The transfer
velocity, $v_t$, depends primarily on meteorological and soil conditions, and is typically on the order of ~1 cm s$^{-1}$. The
flux rate of NO was calculated accordingly with mean daytime NO concentrations of 38 ppt. The calculated flux F*
was about (67±3) % of the flux measured in the chamber.
The distribution of nine different surface cover types was mapped (Fig. 2), including stones, vascular vegetation and
litter not being attributed to emit significant amounts of HONO and NO to the atmosphere. The residual HONO
emitting surface covers comprised 45.6% of total surface in the investigated area. Combining the information on
soil/biocrust population and the calculated flux F*, a site-specific community emission F$_{comm}$ of HONO and NO can
be estimated via following equation (eq. 6).
$$F_{comm,max} = \sum_i^{type} F^*_{max,i} * p_i / 100 \quad\quad \text{or} \quad\quad F_{comm,int} = \sum_i^{type} F^*_{int,i} * p_i / 100 \quad\quad (eq.\ 6)$$
where F$_{comm}$ denotes the estimated community flux, $F^*_{max,i\ or\ int,i}$ the maximum or integrated emission rates of each
individual surface cover type i  [ng N m$^{-2}$ s$^{-1}$ or μg N m$^{-2}$ ] and p$_i$ the fraction of population type i [%].
Under optimum soil water conditions (20-30% WHC) and constant temperatures of about 25°C, between 2.2 and
18.8 ng m$^{-2}$ s$^{-1}$ of total HONO-N and 1.6-16.2 ng m$^{-2}$ s$^{-1}$ of total NO-N are emitted from the different crust/soil
population combinations derived from the vegetation cover assessment. In the lower range of total emissions the
contribution from bare soil dominates with up to 69% (HONO) and 55% (NO), respectively, followed by moss BSC
(HONO: 23%; NO: 32%). At high levels of total emission, the contribution from light BSC dominates (HONO: 43%,
NO: 49%), decreasing the contribution of bare soil down to about 25% (HONO) and 13% (NO). Emissions from
dark BSC contribute about 20% or 24% to the total HONO or NO flux while the contribution from moss BSC
decreases to 10% or 12%, respectively. Emissions from chlorolichen BSC don´t play a significant role (< 2.4%) in
general (see Fig. 7).
After heavy rainfalls moistening the soil to full water-holding capacity, 11-113 µg m$^{-2}$ of HONO-N and 10-131 µg
m$^{-2}$ of NO-N can be calculated for one complete wetting-and-drying period. Assuming 30 rain events per year (based
on the statistic of 4 years precipitation data), a wetting-drying cycle time of 7 days, and constant emissions in
between them (at 10% WHC) up to 160 mg m$^{-2}$ yr$^{-1}$ of nitrogen can be emitted directly by the sum of HONO-N and
NO-N from Cyprus natural ground surfaces, i.e., excluding heterogeneous conversion of NO$_2$ on ground surface.
The release of HONO from the ground surface to the atmosphere can be related to the atmospheric HONO
production rate via eq. 7 (adapted from Su et al., 2011) and then compared to the missing source.

$$S_{ground} = \frac{F * commmax(T,swc)}{BLH} * a \qquad\qquad (eq.7)$$

with $S_{ground}$: HONO or NO emitted from ground surface; BLH: boundary layer height (mixed layer height) and a:
factor to convert ng N in number of molecules ($10^{-9}*6.022 \times 10^{23}/14$).
During the CYPHEX campaign in summer 2014 a mean boundary layer height of 300 m above ground layer was
observed by means of a ceilometer. Due to missing precipitation during CYPHEX, but high relative humidity
prevailing (CYPHEX 2014: 75-100%), a mean soil water content of 10% WHC (at 25°C) can be estimated (Likos,
2008; Leelamanie, 2010), reducing the HONO source strength to 35% of the emission maximum at optimum SWC.
Integrating the lowermost versus the uppermost observed HONO emissions per soil/crust type, the emissions at 25°C
and a SWC of 10% WHC would span a wide range between $1.1 \times 10^5$ and $9.6 \times 10^5$ molecules cm$^{-3}$ s$^{-1}$, covering 9 to
73% of the missing mean source of $1.3 \times 10^6$ molecules cm$^{-3}$ s$^{-1}$ observed in the field (Meusel et al., 2016). However,
temperatures in the field have strong diel cycles, and a temperature increase from 25°C to 50°C has been shown to
lead to 6-10 times higher emission at constant SWC (Oswald et al., 2013; Mamtimin et al., 2016). On Cyprus the
observed soil surface temperatures changed from 10 °C during night up to 45 °C during daytime (Fig. 8, red line, or
Fig. S2). In the natural habitat the micrometeorological parameters change in concert, i.e., with increasing
temperature the SWC decreases, influencing the flux-enhancing effect of temperature. Based on the assumption of a
linear change of SWC with temperature a diel course of the SWC between 6 and 14% of WHC is simulated (Fig. 8,
blue line), lowering the emission flux (22-49% of optimum). Applying the described SWC dependence and the
temperature dependence on flux rates as reported by Oswald et al. (2013), high daytime temperatures increase the
simulated diel course of HONO-N flux up to daytime maximum of 7.4 ng m$^{-2}$ s$^{-1}$ (Fig. 8, lower panel), but with a
notable dip at high noon, due to the opposing effect of decreasing SWC at higher temperatures. The NO-N emissions
show a similar pattern, with a slightly lower flux range (up to 6.4 ng m$^{-2}$ s$^{-1}$). Converted into production rates (eq. 7),
the ground based soil and biocrust emissions at noon would be up to $1.1 \times 10^6$ molecules cm$^{-3}$ s$^{-1}$ HONO-N and 0.9 x
$10^6$ molecules cm$^{-3}$ s$^{-1}$ NO-N  covering up to 85% and 8.5% of the missing HONO and NO source found during
CYPHEX 2014 (Meusel et al., 2016). Note that during CYPHEX there were two periods with lower rH, in which
even a NO sink was detected.  In some mornings of the campaign dew formation was expected causing an increase in
soil humidity. Combined with rising temperatures after sun-rise these optimized meteorological conditions may have
led to enhanced soil emissions and would confer a reasonable explanation for the strong HONO morning peaks
observed during the campaign.

Field observations (VandenBoer et al., 2013; Zhang et al., 2009; Tsai et al. 2017) as well as model results (Wong et al., 2013) showed that HONO concentrations typically decrease exponentially from the surface upwards. Eq. 7 does not include a chemistry-transport model, nor accounts for the existence of a vertical profile of concentrations, which may bias the calculation on HONO source strength. But the method for predicting the ground source using homogeneous mixed air columns is consistent with other recent studies (Stemmler et al., 2006; Tsai et al., 2017). Tsai et al. (2017) clearly showed the presence of an important ground source of daytime HONO at a rural basin in Utah, during wintertime (no snow, low temperatures). They inferred that ground surface fluxes may account for $63\pm32\%$ of the unidentified HONO daytime source throughout the day. HONO-N fluxes of up to 7.4 ng m$^{-2}$ s$^{-1}$ (Fig. 8, lower panel) determined in this study are comparable to HONO-N fluxes found in other regions, e.g., 2.7 ng m$^{-2}$ s$^{-1}$ reported for the northern Michigan forest canopy (Zhang et al., 2009; Zhou et al., 2011), the average daytime HONO-N flux of 7.0 ng m$^{-2}$ s$^{-1}$ measured over an agricultural field in Bakersfield (Ren et al., 2011), and the average HONO-N flux of about 11.6 ng m$^{-2}$ s$^{-1}$) measured by Tsai et al. (2017). In contrast to the present study, the latter concluded that, under the prevailing high NOx conditions, the respective HONO formation was related to solar radiation and NO$_2$ mixing ratios, such as photo-enhanced conversion of NO$_2$ or nitrate photolysis on the ground. This can be ruled out in this study, as pure air (no NO$_2$) was used to purge the chambers and no light was applied. While in Cyprus the observed soil emissions can explain high amounts of atmospheric HONO, other studies excluded soil emission to be a dominant source (Oswald et al., 2015; Laufs et al., 2017). Oswald et al. (2015) studied soil samples from a boreal forest in Finland and observed HONO emission below the detection limit. But those samples had very low nutrient contents and were highly acidic (pH $\approx$ 3) for which microbial activity is supposed to be low (Fierer and Jackson, 2006; Persson and Wiren, 1995; Ste-Marie and Pare, 1999; Simek and Cooper, 2002). Similarly, Laufs et al. (2017) didn´t find correlations between HONO fluxes and temperature or humidity measured in the field, and concluded that other HONO sources than biological soil emission must have been dominated. In contrast to the soil water content in Cyprus, the water contents at the field site studied by Laufs et al. (2016) were higher than the optimum soil water content presented by Oswald et al. (2013).

**4 Conclusions**

HONO and NO emission rates from soil and biological soil crusts were derived by means of lab-based enclosure trace gas exchange measurements, and revealed quite similar ranges of reactive nitrogen source strengths. Emissions of both compounds strongly correlated with NO$_2^-$ and NO$_3^-$ content of the samples. Emissions from bare soil were highest, but bare soil surface spots were rarely found at the investigated CYPHEX field study site. The estimated total ground surface HONO flux in the natural habitat was consistent with the previously unaccounted source estimated for Cyprus, i.e., the unaccounted HONO source can essentially be explained by emissions from soil/biocrusts. For NO, the measured and simulated fluxes cannot account for the unaccounted NO source (during the humid periods of the CYPHEX campaign 2014), indicating that emission from soil was not the only missing source of NO.

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

**Table 1: Overview on the samples, distribution of replicates of soil/biocrust type and the different analysis:**

| Type | Only nutrient analysis | Flux measurements, followed by nutrient and chlorophyll analysis | Sum |
|---|---|---|---|
| Bare soil | 3 | 3 | 6 |
| Dark BSC | 3 | 5 | 8 |
| Light BSC | 3 | 4 | 10 |
| Light BSC + cyanolichen | 3 | | |
| Chlorolichen BSC I | 3 | 3 | 12 |
| Chlorolichen BSC II | | 6 | |
| Moss BSC | 3 | 4 | 7 |
| Sum | 18 | 25 | 43 |

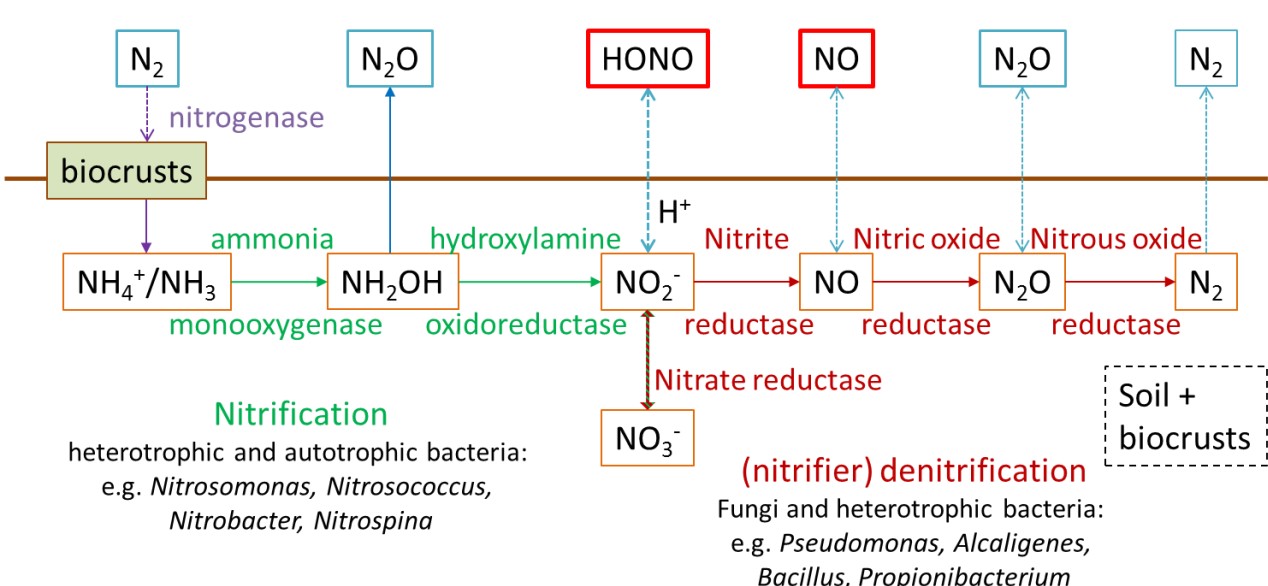

**Fig. 1: Nitrogen cycle at the atmosphere and pedosphere/biosphere interface including nitrogen fixation, nitrification,**
**denitrification and emission. Involved enzymes and organisms are specified.**

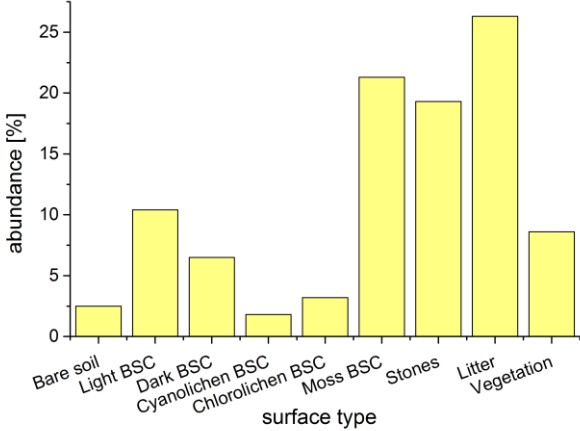

**Fig. 2: Distribution of different types of ground surfaces in the studied area. Information derived from 50 grids.**

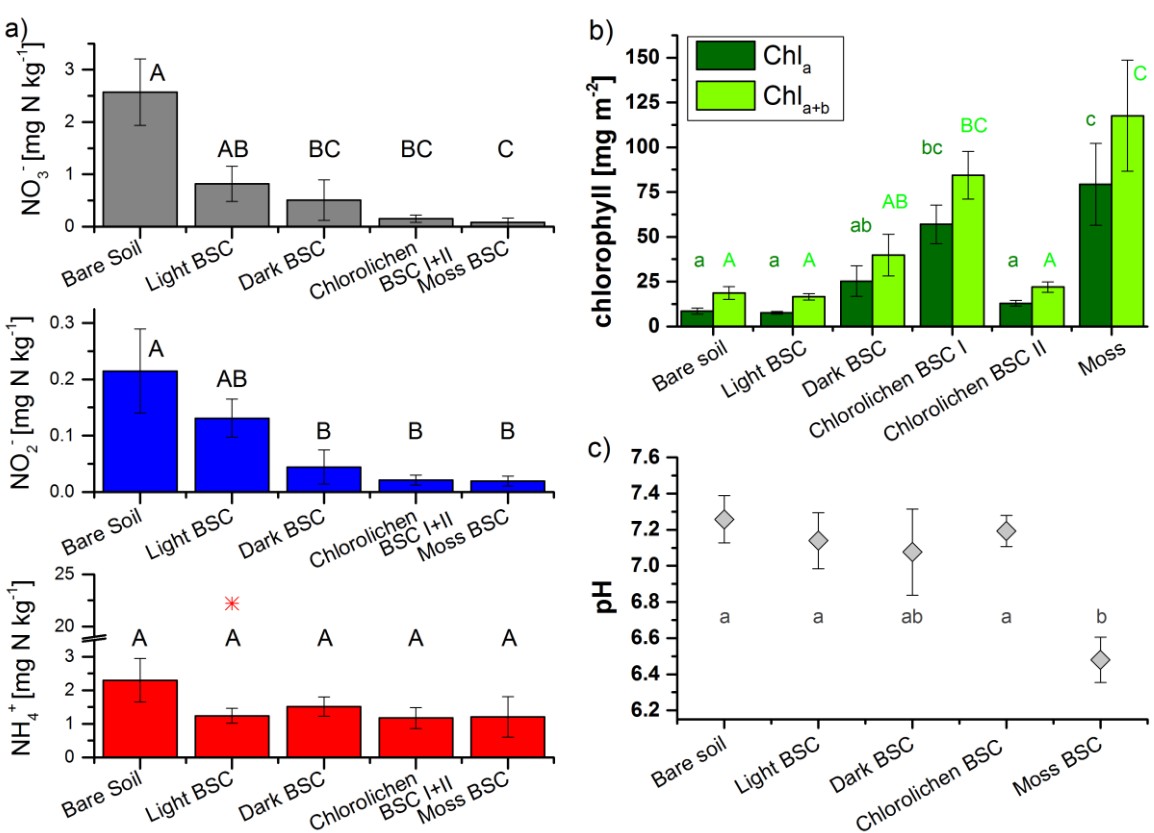

**Fig. 3: Nutrient- and chlorophyll contents as well as pH values of bare soil and biocrust samples of different types. a)**
**Nitrate, nitrite and ammonium content of all replicates. The red star indicates an outlier, b) chlorophyll a and chlorophyll**
**a+b contents of samples after flux measurements c) pH values of samples without and after flux measurements (bare soil**
**and moss BSC: n = 4; light, dark and chlorolichen BSC: n = 3). Number of replicates for a and b see table 1. In all 3 plots**
**error bars indicate standard error of the mean and different letters indicate significant differences (of log-transformed**
**data; p=0.05).**

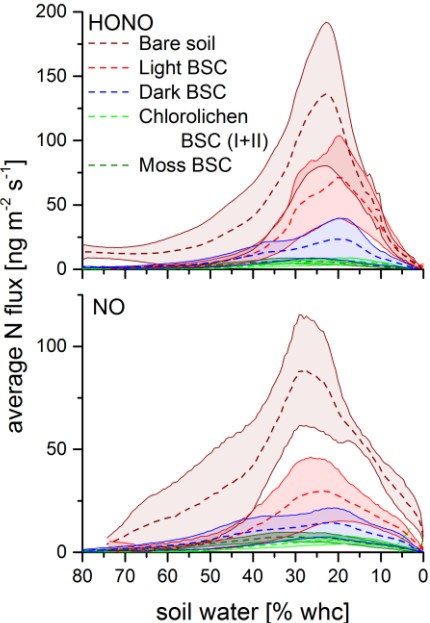

**Fig. 4: HONO and NO emission fluxes as a function of soil water content. Dotted lines are the mean fluxes. Shaded areas**
**indicate the standard deviation.**

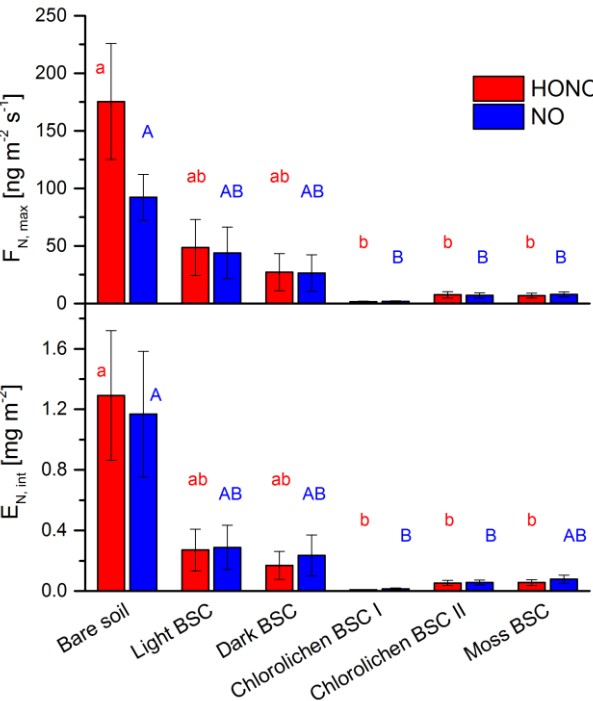

**Fig. 5: Emission of HONO and NO from bare soil and biocrusts. Upper panel: Maximum HONO-N and NO-N fluxes in ng**
**m$^{-2}$ s$^{-1}$ at optimum water conditions; Lower panel: Emissions integrated over a whole wetting-and-drying cycle in mg (N)**
**m$^{-2}$; letters show significant difference (p=0.05,of log-transformed data); error bars indicate standard error of the mean of**
**replicates (bare soil n=3; light BSC n=4; dark BSC n=5; chlorolichen BSC I n=3; chlorolichen BSC II n=6; moss BSC**
**n=4).**

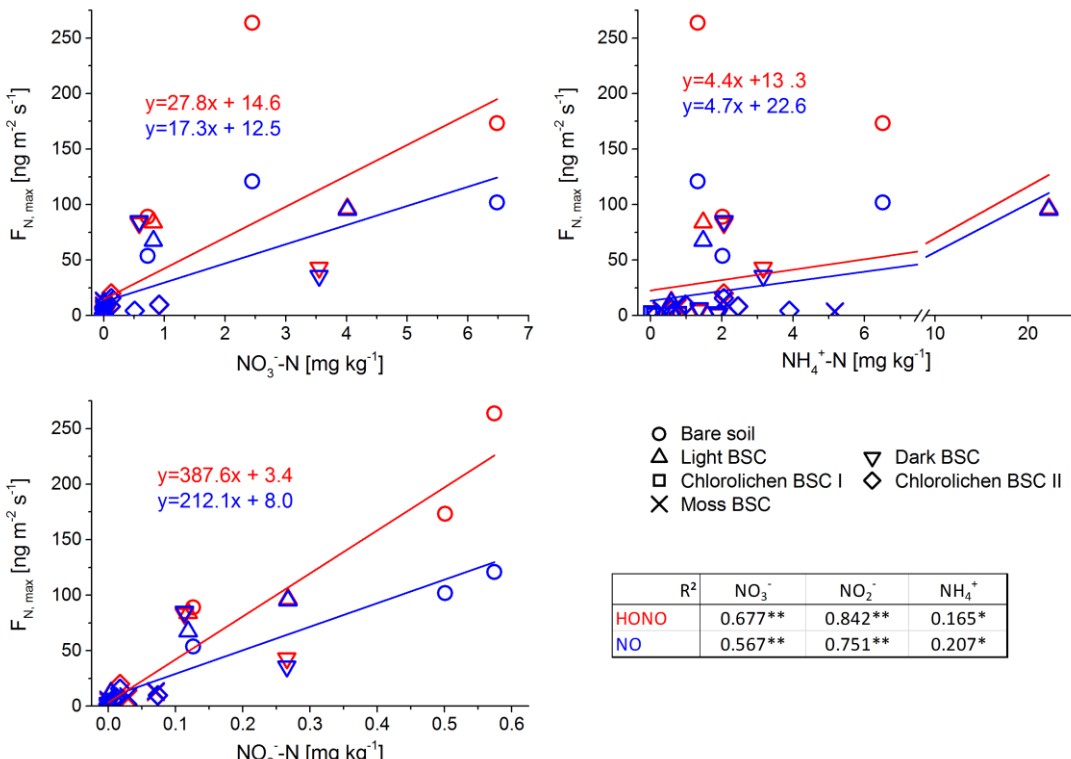

2   **Fig. 6: Correlation between maximum flux of HONO and NO and nutrient content of all Cyprus soil and biocrust samples**
3   **with Pearson correlation factors (of log transformed data; \*\*: p < 0.001; \*: p < 0.05).**

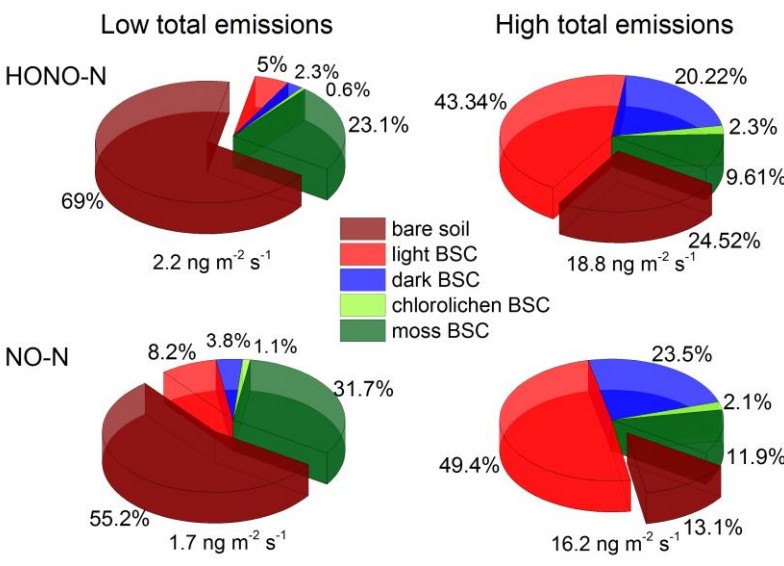

5   **Fig. 7: Contributions of different ground surfaces to the total F\*.**

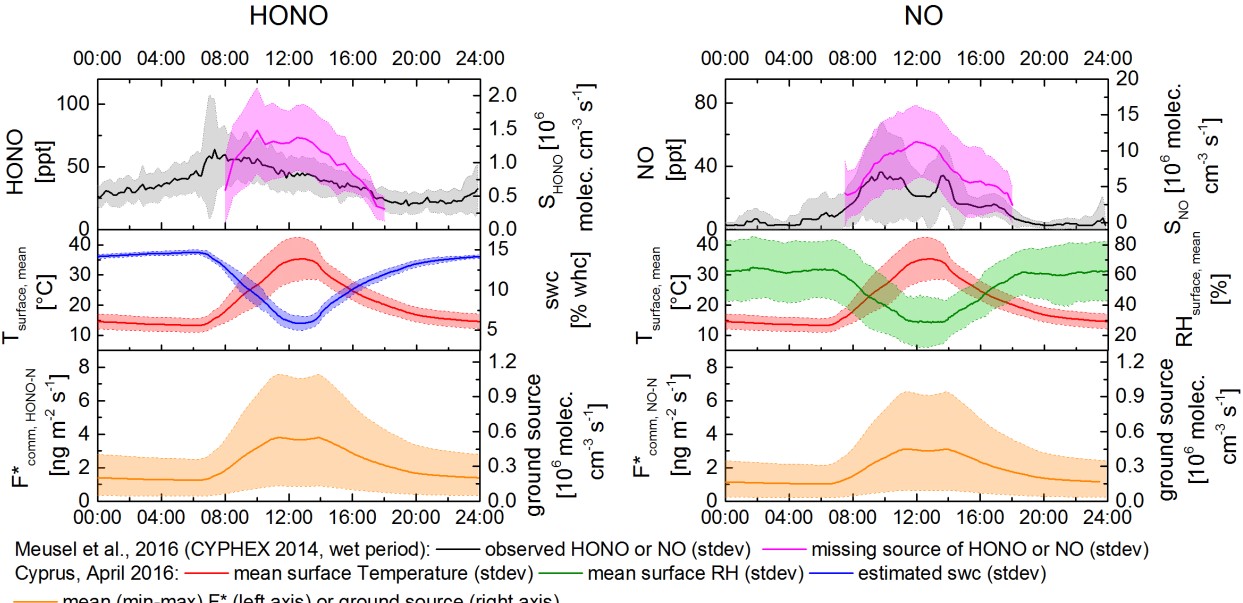

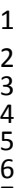

**Fig. 8: Diel pattern for HONO and NO emission in comparison with the observed HONO concentrations and missing source during the CYPHEX 2014 campaign. Upper panels: observed concentration of HONO and NO shown in black, missing source shown in pink. Middle panels: mean surface temperature and mean surface humidity measured in April 2016 in Cyprus and estimated soil water content shown in red, green and blue, respectively. Lower panel: calculated mean F\* (mean temperature) with the area indicating the lower and upper limit.**