# Peer review of "Emission of nitrous acid from soil and biological soil crusts"

_Atmospheric Chemistry and Physics, 2017_

## Referee Comment (RC1) · Anonymous Referee #1 · 9 Jun 2017

In this manuscript, the authors presented laboratory-determined emission rates of HONO and NO from soil and biological soil crust samples collected from arid and semi-arid environments in Cyprus, and extrapolated the results to the ambient conditions. The data and results presented are useful and are suitable for publication in Atmospheric Chemistry and Physics. However, the authors need to address the following comments before I could recommend the acceptance of this manuscript for publication:

General comments:

I am concerned about the validity of extrapolating the laboratory results to the ambi-

ent conditions in this study. The soil and biological soil crust samples were stored at room temperature for up to 15 weeks before some of the experiments were conducted. The samples might be deteriorated during the storage, and by the time of experiments in the laboratory, their chemical (nitrite) and biological (chlorophyll and microbial population) characteristics might be quite different from those under ambient condition. Furthermore, the laboratory experimental conditions were very different from those of the ambient, e.g., air and soil temperature, humidity, and their daily cycles. And finally, while the soil was always a HONO source in the laboratory dynamic chamber since dry zero air was flowing over the soil sample, it could be a net sink for HONO in the air under ambient conditions, for example, during the morning hours when RH is high and a significant level of HONO is present.

While there is no doubt that HONO emission from soils could be an important source of atmospheric HONO under certain conditions, the results from this study should be considered as qualitative, and the actual contribution need to be verified and determined by field studies including flux measurements under ambient conditions. Two recent such measurements suggest that soil emission was not be a significant HONO source in boreal forest (Oswald et al., 2015) and at agricultural field site (Laufs et al., 2017).

I would suggest the authors to add a figure to show diurnal plots of surface temperature and RH (from Figures 2C and 2D), extrapolated HONO and NO emission rates (from Figures 3 and 5, and RH information), and HONO and NO concentrations measured during the CYPHEX field study. Comparison of the diurnal variation patterns of extrapolated HONO flux and ambient HONO concentration should provide us with some insight into the potential importance of soil HONO emission as a HONO source over the day.

Specific comments: Page 4, section 2.4 Trace gas exchange measurements: how was a sample placed into the chamber and what was the thickness the sample. The information would help readers in understanding the data presented.

[Figure]

Page 7, section 3.3 NO and HONO flux measurements: Is the unit of ng m-2 s-1 based on the area that a sample (25-35 g) occupied in the field? Or is it based on the area of the sample occupied in the flow chamber? The authors need to explain how these parameters were derived from laboratory results, even if the method has been discussed in previous papers by the authors.

Figure 5: Would the flux behavior be the same if the experiment is done reversely, i.e., flowing humid air over dry soil. This information may be important to understand if soil HONO emission is important HONO source in the evening and night.

References Oswald, R., et al., A comparison of HONO budgets for two measurement heights at a field station within the boreal forest in Finland. Atmos. Chem. Phys., 15, 799-813, 2015. Laufs, S., et al., Diurnal fluxes of HONO above a crop rotation, Atmos. Chem. Phys. Discuss., https://doi.org/10.5194/acp-2016-1030, in press, 2017.

---

## Referee Comment (RC2) · Anonymous Referee #2 · 9 Jun 2017

The stated objective of this study is to characterize and quantify direct emissions of HONO and NO from soil samples collected from Cyprus. This is a follow-up paper to a study by the same group aimed at characterizing daytime formation of HONO during a larger field campaign (CYPHEX, summer 2014) in the same region of Cyprus. That study concluded that soil microbial source of HONO and NO may have contributed the measured mixing ratios of these gases.

The present manuscript seeks to make that connection between those emissions and soil by carrying out chamber studies on soil collected at this site. The study site was characterized qualitatively using a gridded transect and visual identification to catego-

rize nine types of ground cover (bare soil, light and dark cyanobacteria, chlorolichen, cyanolichen, moss-dominated, stone, litter, and vascular vegetation/shrub). Six of these soil coverage types were sampled and transported to lab to measure HONO and NO emissions using a dynamic chamber method. In addition, the chlorophyll and nutrient (ammonium, nitrate, and nitrite) levels of those samples were measured. Fluxes of gaseous HONO (measured via LOPAP) and NO (measured by chemluminescence) were found to be highest for bare soil, followed by light and dark biocrusts (Light and Dark BSC), which comprise a combined 2.5, 10, and 6 % of the total ground coverage, respectively. Emissions of HONO and NO were correlated to soil nitrite and nitrate levels (not ammonium or other parameters measured). Flux data along with surface coverage information was used to scale up fluxes in an attempt to estimate the contribution of biogenic soil emissions to the HONO and NO budget determined for the CYPHEX campaign. The conclusion of the paper is that biocrust emissions may close the Cyprus HONO budget. The paper is clear, statistical methods are appropriate and the topic is of interest to the atmospheric science and biogeochemistry communities. I have the following concerns about this manuscript regarding the study's approach, the appropriateness of the laboratory flux approach, and its conclusions.

Sampling methods. Section 2.1 on sampling methods focuses on the procedure used to visually assign and quantify the surface coverage using the grid method, but lacks details on the sampling method used to collect samples for the laboratory chamber study. Details are limited to: "Each sample was collected in a plastic petri dish, sealed and stored in the dark at room temperature until further analysis (storage time less than 15 weeks)." What form did these samples have? What was their dimension and mass? How deep did the samples extend into the ground? Was the sample that was placed into the soil chamber a whole core or was it sieved and/or prepared in any way? The authors state that the storage time in laboratory was less than 15 weeks. Were samples around this long before the nutrient levels were measured, or were nutrient measurements made sooner. Much can happen 15 weeks, and nitrification can be taking place during storage that changes the nutrient pool and impact the lab measurements. This

can contribute to significant variability of certain soil measurements.

The sampling procedure and consistency/physical properties of the sample that was placed in the chambers is critical for this type of study. There has been a debate among researchers about how representative gas fluxes are for sieved or cored soil samples of environmental conditions. Previous studies suggest that such laboratory studies of soil cores give similar flux measurements as eddy covariance for grassland soils. In such soils, the surface porosity can be considered to be more similar to porosity of soil just below the surface and arguments could be made that gas exchange from soil in the field and in laboratory cores might be similar. However, biocrusts may present a particularly difficult biome to sample in this way since the intact soil and disturbed soil may have very different structural properties. The physical structure of these surfaces is defined by a network of filamentous growth and biomass that creates a hard crust that is often an impermeable layer that may impact gas exchange. These structural features are known to form hard crusts that prevent soil erosion in sensitive arid ecosystems. The soil exposed when soil is extracted as a core or sieved soil may provide a means to bypass surface structural properties that hinder gas exchange. Do the authors have any evidence to suggest that their method of sampling did not impact gas exchange from these samples? It is important to demonstrate that the results are close to reality and can be used for the type of scaled up estimation performed at the end of the manuscript.

While the physical appearance of biological soil crusts is a useful classification tool, it does not provide any information on the actual nitrification processes that occur in or below the biocrust and may be responsible for controlling soil emissions of HONO and NO. Much of the molecular biology that is important for atmosphere-land interactions is likely occurring just belowground (i.e., below the crust that is visible at the surface). It is also misleading to focus solely on the moss, lichen, actinobacteria, which are not the direct sources of these gases. Although biocrusts affect nutrient availability via N fixation, it is their possible associations with ammonia (and nitrite) oxidizing microbes

(bacterial and archaea) that ultimately convert the fixed nitrogen to nitrite and nitrate. The current study does not consider the role of ammonia oxidizing microbes in association with biocrusts or the other surface types in the area. These microorganisms are not limited to living within or under biocrusts, but are present in most other soil types to differing degrees. It does make sense that such nitrifying organisms will thrive where their substrates are abundant. However, there are numerous other soil types where this may be the case. Further, there may be many other soil organisms that compete with nitrifiers for their substrates, that may reduce their abundance in soil that would seemingly favor nitrifier populations.

The literature that does exist (e.g., Frontiers in Microbiology 2016, doi: 10.3389/fmicb.2016.00505) on biocrust-nitrifier associations suggests that biocrusts do not necessarily host a greater abundance of ammonia oxidizing organisms compared to soil supporting trees, nitrogen fixing shrubs, etc. This is an important topic to address. Related to this, Figure 3 of the current manuscript demonstrates that there are other soil types throughout the landscape characterized by stones, litter, and vegetation cover that do not have associated flux values and were not included in the final conclusion regarding relative importance of biocrusts in HONO and NO emissions. The model only considered the approximately 45% of the surface types whose fluxes were characterized. It is possible that fluxes in the other soil types had as high or higher fluxes? If so, would this not make the estimate of contributions of soil emissions to overall atmospheric composition higher and possibly overshoot the Cyprus HONO budget determined in the field campaign? Indeed, Figure 8 is somewhat misleading since it must be noted that F* only refers to the total HONO and NO flux associated with the 45% of surface types that were actual studied. It is very possible that the pie charts would look very different if other surfaces types were considered. So there is a large uncertainty here.

In my opinion, a satisfying or conclusive connection between the soil emissions of NO and HONO and biocrusts has not been made. The most one can conclude from

this study is that volatilization from soil bound nitrite could contribute to the NO and HONO measured in the air above the soil. Indeed, it may have been useful for the authors to include a better discussion of why they can rule out long range transport and atmospheric deposition of nitrate and NOx over time as the source of HONO and NO precursors to this soil. Even though this particular area of Cyprus may have a low population, is possible for it to accumulate anthropogenic inputs from population centers surrounding the Mediterranean basin over time? One is left wondering whether the results support the paper's title and the conclusions it suggests.

Lastly, Figure 1 presents a month of meteorological data (air and surface T, air and surface %RH, and precipitation) at the site for the month before samples were taken. The data features prominantly as Figure 2, yet is not used. So, it is unclear why an entire figure was devoted to this data when averages for these values during the time of sampling could have been provided in the text.

In conclusion, I feel that the strengths of this manuscript are that it is mostly well written and provides supporting evidence for the fact that soil emissions could have impacted the NOx and HONO budget during the CYPHEX 2014 field campaign. Weaknesses include: (i) there is minimal evidence from this study to support that the emissions are biological in nature (outside of the fact that the flux vs. soil moisture plot matches those of studies on pure cultures of ammonia oxidizing bacteria, Oswald et al.) and (ii) there is less evidence that the actual biocrusts are the dominant HONO and NO sources in this area since we have no data on emissions from 55% of the other surface types present in the study area. Care must be taken here to not draw too much information from these results. The approach described in this paper is not unique; its novelty is related to providing data on soil HONO and NO emissions from understudied region of the globe. Due to its limited scope, this study would have been better suited as supporting data to include in the field campaign paper by Meusel et al. 2016. It may be possible for this study to stand on its own if the above concerns are appropriately addressed in a revised manuscript.

---

## Referee Comment (RC3) · Anonymous Referee #3 · 10 Jun 2017

Summary: Soil samples used in this work are from soils collected from the field, manipulated in a controlled lab environment, and then measured fluxes extrapolated to compare with the missing HONO source calculated for the CYPHEX field campaign in the same location. Soils were collected and categorized from a gridded sampling scheme. HONO and NO fluxes were measured from the soils in replicates in order to quantify which surface soil community members, if any, were responsible for the majority of the HONO fluxes observed. The authors performed nice controlled experiments in the lab and found some interesting conclusions, counter to previous findings in similar soils by this group. The manuscript may be acceptable for publication in Atmospheric Chemistry and Physics, subject to a number of concerns being addressed.

Major Comments: 1. There is no experimental control of soils devoid of microbial activity for each soil type. One would think it necessary to fumigate (or otherwise kill) soil samples of each type to control for biotic versus abiotic HONO and NO emissions, yet this is not presented. If possible, the authors should consider acquiring this data and adding it to the manuscript for comparison and correction of the dataset.

2. The consideration of the effects of the measured soil pH on HONO release using the method of the Su et al. (2011) work is not considered in the interpretation of the data. What proportion of the emissions measured in each case can be ascribed to simple partitioning? What effect does this have on comparisons between soil types considered in this work when abiotic exchange is estimated versus measured (see comment above) in the experimentally measured fluxes? A major concern is that if abiotic partitioning from dead soils is not favored by calculation from bulk pH measurements and nutrient loadings (i.e. soil pH » pKa HONO) then another emission mechanism is active and should be considered/discussed.

3. Scaling to atmospheric relevance for the field campaign is very interesting, but inappropriate to include in the title of the manuscript. The linkage between microbial activity and HONO and NO emissions is of growing importance to constrain and these lab-based measurements help to do so. However, the uncertainty in the extrapolation of the data to the field campaign observations of the missing daytime HONO source covers an order of magnitude range, which means at the low end of the estimate these processes account for < 10 % of the daytime HONO source. The authors do not discuss this limitation and should do so. The title of the manuscript should also remove the HONO budget closure implications due to this significant uncertainty. An important consideration here is one that has been reported and discussed much in the atmospheric community over the past 5 years and that is the vertical structure of HONO, and by proxy, the daytime HONO source strength near the ground surface. Using soil HONO emissions to scale to measurements made nearly 6 m above the ground may add additional error in the budget closure calculation as the perceived missing source

changes with height. This topic and the relevant references would be a worthwhile addition to this component of the discussion as there are several reports on vertical structure of HONO in arid agricultural and rural regions in the US.

4. The quality of the nutrient data cannot be determined as no accuracy, precision, or detection limit values are presented. The data table in the supplement reports measurements of zero, which should be represented by '< LOD' and the detection limits determined by the experimental runs calculated (do not use the instrument manufacturer's stated values). My concern is that some of the measurements made on the samples are near the detection limits and therefore highly uncertain, which may confound the comparisons made throughout the manuscript. The authors should also be more cautious in their reported values from these measurements. Are these analytically certain to so many significant digits?

5. The experiments performed on nutrient content before and after experiments (Figure 4) is good to have performed, but this data does not need to be presented in the figure and would help make this a cleaner plot. The data can be replaced with one sentence in the manuscript stating that the nutrient levels were not different in the soils by performing the flux experiments. To that end, this suggests that the full set of measurements of these nutrients could be pooled and improve the statistical analyses performed, as it would reduce the standard error in the measurements. Further to this point, where replicate measurements have been made, the authors should be presenting the standard error of the mean and not the standard deviation as it aids in connecting the reader to the statistical results. The results and discussion surrounding the purpose, method selection, and outcomes of these statistical tests needs to be improved either through clarity in existing sections or expansion of the text.

6. Details on linear regression in Figure 7 needs to be presented. There is presumably large uncertainty in both measurements being compared and the appropriate considerations must be included prior to assessing the relationship between them, along with the associated uncertainty in the result. Building on this, the direction of the trend

and also its associated uncertainty from the regression fit may be more telling towards whether the strength of the coefficient is truly robust or being limited by the sample size available.

7. Figure 2 needs to be streamlined to present the relevant information for the contents of the manuscript. The level of detail here is not necessary.

Minor Comments:

Fix tense and plurality issues throughout the manuscript.

Many sentences have issues with comma splicing, making them long and the purpose of the sentence difficult to follow.

---

## Author Comment (AC1) · 9 Oct 2017

**Overview:**
*In this manuscript, the authors presented laboratory-determined emission rates of HONO and NO from soil and biological soil crust samples collected from arid and semi-arid environments in Cyprus, and extrapolated the results to the ambient conditions. The data and results presented are useful and are suitable for publication in Atmospheric Chemistry and Physics. However, the authors need to address the following comments before I could recommend the acceptance of this manuscript for publication:*

**Comment:**
*I am concerned about the validity of extrapolating the laboratory results to the ambient conditions in this study. The soil and biological soil crust samples were stored at room temperature for up to 15 weeks before some of the experiments were conducted. The samples might be deteriorated during the storage, and by the time of experiments in the laboratory, their chemical (nitrite) and biological (chlorophyll and microbial population) characteristics might be quite different from those under ambient condition.*

**Response:**
This is a very good and valid comment. Some of the authors (Dianming Wu, Alexandra Tamm, Bettina Weber) have already investigated general storage properties of biological soil crust and soil samples and their measurements showed no significant loss of chlorophyll or nitrogen compounds during 4-month storage at room temperature as long as the samples were stored dry and in the dark. Also the storage temperature (-20°C, 4°C or room temperature) had no significant effect on nutrient/chlorophyll/fluxes. The respective study will be submitted soon.
A short note was added in the manuscript (chapter 2.1):
"Based on previous experiments in our laboratory, it can be anticipated that the sample's chemical (nutrient content) and biological (chlorophyll content) properties were not deteriorated during storage (a manuscript on this study will be submitted soon)."

**Comment:**
*Furthermore, the laboratory experimental conditions were very different from those of the ambient, e.g., air and soil temperature, humidity, and their daily cycles. And finally, while the soil was always a HONO source in the laboratory dynamic chamber since dry zero air was flowing over the soil sample, it could be a net sink for HONO in the air under ambient conditions, for example, during the morning hours when RH is high and a significant level of HONO is present.*

**Response:**
This is a good point. Of course, in the field soil temperature varies a lot following the solar radiation, ranging from 15° to 50°C. In the lab the average ambient temperature of 25°C was chosen and kept constant. Soil humidity probably also changes following diurnal cycles of ambient RH and temperature, but during the CYPHEX campaign soil humidity was very low caused by missing precipitation events.
As shown in Su et al. (2011) and VandenBoer et al. (2015), the soil could serve either as a source or as a sink depending on the difference between the equilibrium concentration at soil surface [HONO]* and the ambient [HONO]. Thus, instead of using the flux measured in the lab, we calculated [HONO]*, compared it to the ambient concentration, and then determined the flux ($F^* = v_T \cdot ([HONO]^* - [HONO])$). The soil would act as a sink

when ambient concentrations were higher than [HONO]*. This was not the case, so the soil acted as a source (during daytime).

Please see also our comment below on diel patterns.

**Comment:**
*While there is no doubt that HONO emission from soils could be an important source of atmospheric HONO under certain conditions, the results from this study should be considered as qualitative, and the actual contribution need to be verified and determined by field studies including flux measurements under ambient conditions. Two recent such measurements suggest that soil emission was not be a significant HONO source in boreal forest (Oswald et al., 2015) and at agricultural field site (Laufs et al., 2017).*

**Response:**
We need to point out here, that in both studies mentioned above the ecosystems were very different to the one investigated in our study. While we studied samples from a Mediterranean dryland habitat, a boreal forest (Oswald et al., 2015) and an agricultural field (Laufs et al., 2017) were investigated in the other studies.

Oswald et al. (2015) measured HONO concentrations at two different heights, observed positive and negative gradients but fluxes were not determined. Laufs et al. (2017) also determined the flux and found positive fluxes during daytime. But both studies excluded soil emission to be a major source of HONO.

In dynamic chamber experiments Oswald et al. (2015) measured the HONO and NO flux and found emissions lower than or around the detection limit of 0.08 or 1 ng $m^{-2}$ $s^{-1}$. But the forest soils from Finland had much lower nutrient contents compared to our study. Furthermore, a very low pH of 3 was found, at which a low diversity of soil bacteria was observed (Fierer and Jackson, 2006) and most bacteria won´t be active. It was shown that nitrification rates are very low at pH < 4 (Persson and Wiren, 1995; Ste-Marie and Pare, 1999) while it is not so clear for denitrification rates (Simek and Cooper, 2002).

Laufs et al. (2017) only indirectly excluded biological emission, as the soil in the field had higher soil water contents than the optimum soil water content found in Oswald et al. (2013). Furthermore, they didn´t find a significant correlation to temperature or humidity, what would be expected from biological soil emission. Instead they detected a positive correlation between the HONO flux and $NO_2*J$.

In our samples nutrient content was high and chamber studies showed a good correlation of HONO and NO fluxes to nutrient content. The soil humidity in the field can be assumed to be about 10% whc (as was observed for soils at high relative humidity; see Likos (2008) and Leelamanie (2010)). Thus, we consider our measurements, results and interpretations as being reasonable.

Moreover, Wong et al. (2013) also demonstrated that beside flux measurements, the $HONO/NO_2$ ratio can also be used to identify a surface source or to distinguish between surface and atmospheric source, respectively. A surface source results in a more pronounced diel pattern of $HONO/NO_2$ (with a peak around noon) while an atmospheric source (aerosol surface reactions) leads to a near constant $HONO/NO_2$. During the CYPHEX campaign a clear diel pattern of $HONO/NO_2$ was observed (Fig. R1; Meusel et al., 2016), indicating a surface source. As heterogeneous $NO_2$ conversion was supposed to play a minor role in HONO formation (NOx levels were too low) the soil emission is likely the major source of HONO in Cyprus.

[Figure]

Fig R1: Mean diel course of HONO/NO2 during CYPHEX.

In the revised version of the manuscript (end of chapter 3.4) we now added: "While in Cyprus the observed soil emissions can explain high amounts of atmospheric HONO, other studies excluded soil emission to be a dominant source (Oswald et al., 2015; Laufs et al., 2017). Oswald et al. (2015) studied soil samples from a boreal forest in Finland and observed HONO emission below the detection limit. But those samples had very low nutrient contents and were highly acidic (pH ≈ 3) for which microbial activity is supposed to be low (Fierer and Jackson, 2006; Persson and Wiren, 1995; Ste-Marie and Pare, 1999; Simek and Cooper, 2002). Similarly, Laufs et al. (2017) didn´t find correlations between HONO fluxes and temperature or humidity measured in the field, and concluded that other HONO sources than biological soil emission must have been dominated. In contrast to the soil water content in Cyprus, the water contents at the field site studied by Laufs et al. (2017) were higher than the optimum soil water content presented by Oswald et al. (2013)."

**Comment:**
*I would suggest the authors to add a figure to show diurnal plots of surface temperature and RH (from Figures 2C and 2D), extrapolated HONO and NO emission rates (from Figures 3 and 5, and RH information), and HONO and NO concentrations measured during the CYPHEX field study. Comparison of the diurnal variation patterns of extrapolated HONO flux and ambient HONO concentration should provide us with some insight into the potential importance of soil HONO emission as a HONO source over the day.*

**Response:**
Following the reviewer's suggestion, we estimated a diel pattern of HONO fluxes and included a new figure. When using the same correlation between HONO or NO flux and temperature as found by Oswald et al., (2013) and assuming a slight/linear diel change of soil water content at higher temperature we estimate the following diel pattern for the HONO and NO emission (Fig. R2). For a mean surface temperature ranging from 15-35°C we estimated a soil water content varying between 14 and 6 % whc (average 10% whc as described by Likos (2008) and Leelamanie (2010) for high ambient relative humidity) leading to emissions between 49 and 22% of the optimum flux. The HONO-N flux ranges from 0.5 to 7.5 ng $m^{-2}$ $s^{-1}$ (for the mean temperature; indicated by the orange area in the figure, left lower panel). With rising temperatures and a concurrent drop in swc, the flux increase, but has a small dip around noon. As already indicated in the original manuscript, we can convert the emission flux into a ground based source. Around noon, emissions explain about 70% of the missing HONO source. Similarly, NO emission and sources can be calculated. They are slightly lower than the HONO emission.

[Figure]

Fig. R2 (Fig. 8 in the revised manuscript): Diel pattern for HONO and NO emission in comparison with the observed HONO concentrations and missing source during the CYPHEX 2014 campaign. Upper panels: observed concentration of HONO and NO shown in black, missing source shown in pink. Middle panels: mean surface temperature and mean surface humidity measured in April 2016 in Cyprus and estimated soil water

content shown in red, green and blue, respectively. Lower panel: calculated mean F* (mean temperature) with the area indicating the lower and upper limit.

Instead of only showing the "one-point-study" in the manuscript, we changed the respective text section presenting the following thoughts and add the new figure into the manuscript (after Eq. 7):

"During the CYPHEX campaign in summer 2014 a mean boundary layer height of 300 m above ground layer was observed by means of a ceilometer. Due to missing precipitation during CYPHEX, but high relative humidity prevailing (CYPHEX 2014: 75-100%), a mean soil water content of 10% whc (at 25°C) can be estimated (Likos, 2008; Leelamanie, 2010), reducing the HONO source strength to 35% of the emission maximum at optimum swc. Integrating the lowermost versus the uppermost observed HONO emissions per soil/crust type, the emissions at 25°C and a swc of 10% whc would span a wide range between $1.1 \times 10^5$ and $9.6 \times 10^5$ cm$^{-3}$ s$^{-1}$, covering 9 to 73% of the missing mean source of $1.3 \times 10^6$ cm$^{-3}$ s$^{-1}$ observed in the field (Meusel et al., 2016). However, temperatures in the field have strong diel cycles, and a temperature increase from 25°C to 50°C has been shown to lead to 6-10 times higher emission at constant swc (Oswald et al., 2013; Mamtimin et al., 2016). On Cyprus the observed soil surface temperatures changed from 10 °C during night up to 45 °C during daytime (Fig. 8, red line, or Fig. S2). In the natural habitat the micrometeorological parameters change in concert, i.e., with increasing temperature the swc decreases, influencing the flux-enhancing effect of temperature. Based on the assumption of a linear change of swc with temperature a diel course of the swc between 6 and 14% of whc is simulated (Fig. 8, blue line), lowering the emission flux (22-49% of optimum). Applying the described swc dependence and the temperature dependence on flux rates as reported by Oswald et al. (2013), high daytime temperatures increase the simulated diel course of HONO-N flux up to daytime maximum of 7.5 ng m$^{-2}$ s$^{-1}$ (Fig. 8, lower panel), but with a notable dip at high noon, due to the opposing effect of decreasing swc at higher temperatures. The NO-N emissions show a similar pattern, with a slightly lower flux range (up to 6.4 ng m$^{-2}$ s$^{-1}$). Converted into production rates (Eq. 7), the ground based soil and biocrust emissions at noon would be up to $1.1 \times 10^6$ cm$^{-3}$ s$^{-1}$ HONO-N and $0.9 \times 10^6$ cm$^{-3}$ s$^{-1}$ NO-N covering up to 85% and 8.5% of the missing HONO and NO source found during CYPHEX 2014 (Meusel et al., 2016)."

**Specific comment:**
*Page 4, section 2.4 Trace gas exchange measurements: how was a sample placed into the chamber and what was the thickness the sample. The information would help readers in understanding the data presented.*

**Response:**
The soil/biocrust samples were located in plastic petri dishes measuring 5.5 cm in diameter and about 1 cm in height. The sample to be measured was watered to full whc and the chamber was opened to place the sample in the center on the bottom of the chamber.
Besides the dimensions of the sample this was already described in the manuscript. Dimensions are now additionally described in the revised version of the manuscript.

**Specific comment:**
*Page 7, section 3.3 NO and HONO flux measurements: Is the unit of ng m-2 s-1 based on the area that a sample (25-35 g) occupied in the field? Or is it based on the area of the sample occupied in the flow chamber? The authors need to explain how these parameters were derived from laboratory results, even if the method has been discussed in previous papers by the authors.*

**Response:**
The samples were taken in the same petri dish that was placed into the chamber. So 25-35 g soil have a geometric surface of 23.8 cm² (petri dish). Fluxes were calculated for 1 m². Calculations are now explained in more detail in the supplement:
"Calculations of fluxes derived by dynamic chamber measurements:

$$[HONO] \cdot \frac{f}{A} \cdot \frac{p}{R \cdot T} \cdot M_N = F_{HONO-N} \qquad \text{(eq. S1)}$$

$$[NO] \cdot \frac{fr}{A} \cdot \frac{p}{R \cdot T} \cdot M_N = F_{NO-N} \qquad \text{(eq. S2)}$$

[HONO], [NO] measured mixing ratios in ppb
f = flow rate in $m^3 \ s^{-1}$ (8 L $min^{-1}$ = 1.33 x $10^{-4} \ m^3 \ s^{-1}$)
A = surface of sample in $m^2$ (0.00238 $m^2$)
p = pressure in Pa
R = ideal gas constant (8.31 J $mol^{-1} \ K^{-1}$)
T = temperature in K (298 K)
$M_N$ = molar weight of N (14 g $mol^{-1}$)
$F_{HONO-N}$, $F_{NO-N}$ = fluxes of HONO-N and NO-N in ng $m^{-2} \ s^{-1}$ "

**Specific comment:**

*Figure 5: Would the flux behavior be the same if the experiment is done reversely, i.e., flowing humid air over dry soil. This information may be important to understand if soil HONO emission is important HONO source in the evening and night.*

**Response:**

We performed such an experiment and also observed HONO emission from dry soil flushed with humidified air (see Fig. R3). But we didn´t quantify HONO emissions over a wide range of humidity, yet. In near future we want to study this in more detail.

[Figure]

Fig. R3: HONO and NO emission from dry soil flushed with humidified air (rH ~85%). In this experiment soil from a local field around Mainz, Germany was taken, which was probably fertilized some time before sampling. The soil was coated on a glass tube (30 cm length, i.d. 0.9 cm, soil layer ~ 1 mm) according to Li et al., 2016. The gray bar indicates the time period when the coated flow tube introduced (at 0 min) and eliminated (at 105 min) from the gas exchange system.

**References**

Fierer, N., and Jackson, R. B.: The diversity and biogeography of soil bacterial communities, Proceedings of the National Academy of Sciences of the United States of America, 103, 626-631, 10.1073/pnas.0507535103, 2006.

Li, G., Su, H., Li, X., Kuhn, U., Meusel, H., Hoffmann, T., Ammann, M., Pöschl, U., Shao, M., and Cheng, Y.: Uptake of gaseous formaldehyde by soil surfaces: a combination of adsorption/desorption equilibrium and chemical reactions, Atmos. Chem. Phys. Discuss., 2016, 1-29, 10.5194/acp-2016-273, 2016.

Laufs, S., et al., Diurnal fluxes of HONO above a crop rotation, Atmos. Chem. Phys. Discuss., 10.5194/acp-2016-1030, in press, 2017.

Leelamanie, D. A. L.: Changes in Soil Water Content with Ambient Relative Humidity in Relation to the Organic Matter and Clay. , Tropical Agricultural Research and Extension, 13, 6-10, 10.4038/tare.v13i1.3130, 2010.

Likos, W. J.: Vapor adsorption index for expansive soil classification, Journal of Geotechnical and Geoenvironmental Engineering, 134, 1005-1009, 10.1061/(asce)1090-0241(2008)134:7(1005), 2008.

Mamtimin, B., Meixner, F. X., Behrendt, T., Badawy, M., and Wagner, T.: The contribution of soil biogenic NO and HONO emissions from a managed hyperarid ecosystem to the regional NOx emissions during growing season, Atmos. Chem. Phys., 16, 10175-10194, 10.5194/acp-16-10175-2016, 2016.

Meusel, H., Kuhn, U., Reiffs, A., Mallik, C., Harder, H., Martinez, M., Schuladen, J., Bohn, B., Parchatka, U., Crowley, J. N., Fischer, H., Tomsche, L., Novelli, A., Hoffmann, T., Janssen, R. H. H., Hartogensis, O., Pikridas, M., Vrekoussis, M., Bourtsoukidis, E., Weber, B., Lelieveld, J., Williams, J., Pöschl, U., Cheng, Y., and Su, H.: Daytime formation of nitrous acid at a coastal remote site in Cyprus indicating a common ground source of atmospheric HONO and NO, Atmos. Chem. Phys., 16, 14475-14493, 10.5194/acp-16-14475-2016, 2016.

Oswald, R., et al., A comparison of HONO budgets for two measurement heights at a field station within the boreal forest in Finland. Atmos. Chem. Phys., 15, 799-813, 2015.

Oswald, R., Behrendt, T., Ermel, M., Wu, D., Su, H., Cheng, Y., Breuninger, C., Moravek, A., Mougin, E., Delon, C., Loubet, B., Pommerening-Roeser, A., Soergel, M., Poeschl, U., Hoffmann, T., Andreae, M. O., Meixner, F. X., and Trebs, I.: HONO Emissions from Soil Bacteria as a Major Source of Atmospheric Reactive Nitrogen, Science, 341, 1233-1235, 10.1126/science.1242266, 2013.

Persson, T., and Wirén, A.: Nitrogen mineralization and potential nitrification at different depths in acid forest soils, Plant and Soil, 168, 55-65, 10.1007/bf00029313, 1995.

ŠImek, M., and Cooper, J. E.: The influence of soil pH on denitrification: progress towards the understanding of this interaction over the last 50 years, European Journal of Soil Science, 53, 345-354, 10.1046/j.1365-2389.2002.00461.x, 2002.

Ste-Marie, C., and Paré, D.: Soil, pH and N availability effects on net nitrification in the forest floors of a range of boreal forest stands, Soil Biology and Biochemistry, 31, 1579-1589, 10.1016/S0038-0717(99)00086-3, 1999.

Su, H., Cheng, Y., Oswald, R., Behrendt, T., Trebs, I., Meixner, F. X., Andreae, M. O., Cheng, P., Zhang, Y., and Poeschl, U.: Soil Nitrite as a Source of Atmospheric HONO and OH Radicals, Science, 333, 1616-1618, 10.1126/science.1207687, 2011.

VandenBoer, T. C., Young, C. J., Talukdar, R. K., Markovic, M. Z., Brown, S. S., Roberts, J. M., and Murphy, J. G.: Nocturnal loss and daytime source of nitrous acid through reactive uptake and displacement, Nature Geosci, 8, 55-60, 10.1038/ngeo2298, 2015.

Wong, K. W., Tsai, C., Lefer, B., Grossberg, N., and Stutz, J.: Modeling of daytime HONO vertical gradients during SHARP 2009, Atmospheric Chemistry and Physics, 13, 3587-3601, 10.5194/acp-13-3587-2013, 2013.

---

## Author Comment (AC2) · 9 Oct 2017

**Overview:**

*The stated objective of this study is to characterize and quantify direct emissions of HONO and NO from soil samples collected from Cyprus. This is a follow-up paper to a study by the same group aimed at characterizing daytime formation of HONO during a larger field campaign (CYPHEX, summer 2014) in the same region of Cyprus. That study concluded that soil microbial source of HONO and NO may have contributed the measured mixing ratios of these gases. The present manuscript seeks to make that connection between those emissions and soil by carrying out chamber studies on soil collected at this site. The study site was characterized qualitatively using a gridded transect and visual identification to categorize nine types of ground cover (bare soil, light and dark cyanobacteria, chlorolichen, cyanolichen, moss-dominated, stone, litter, and vascular vegetation/shrub). Six of these soil coverage types were sampled and transported to lab to measure HONO and NO emissions using a dynamic chamber method. In addition, the chlorophyll and nutrient (ammonium, nitrate, and nitrite) levels of those samples were measured. Fluxes of gaseous HONO (measured via LOPAP) and NO (measured by chemluminescence) were found to be highest for bare soil, followed by light and dark biocrusts (Light and Dark BSC), which comprise a combined 2.5, 10, and 6 % of the total ground coverage, respectively. Emissions of HONO and NO were correlated to soil nitrite and nitrate levels (not ammonium or other parameters measured). Flux data along with surface coverage information was used to scale up fluxes in an attempt to estimate the contribution of biogenic soil emissions to the HONO and NO budget determined for the CYPHEX campaign. The conclusion of the paper is that biocrust emissions may close the Cyprus HONO budget.*

*The paper is clear, statistical methods are appropriate and the topic is of interest to the atmospheric science and biogeochemistry communities.*
*I have the following concerns about this manuscript regarding the study's approach, the appropriateness of the laboratory flux approach, and its conclusions.*

**Comment:**

***Sampling methods***. *Section 2.1 on sampling methods focuses on the procedure used to visually assign and quantify the surface coverage using the grid method, but lacks details on the sampling method used to collect samples for the laboratory chamber study. Details are limited to: "Each sample was collected in a plastic petri dish, sealed and stored in the dark at room temperature until further analysis (storage time less than 15 weeks)." What form did these samples have? What was their dimension and mass? How deep did the samples extend into the ground? Was the sample that was placed into the soil chamber a whole core or was it sieved and/or prepared in any way? The authors state that the storage time in laboratory was less than 15 weeks. Were samples around this long before the nutrient levels were measured, or were nutrient measurements made sooner. Much can happen 15 weeks, and nitrification can be taking place during storage that changes the nutrient pool and impact the lab measurements. This can contribute to significant variability of certain soil measurements.*

**Response:**

In order to take a sample in the field, the bottom part of a plastic petridish (diameter: 5.5 cm, height: 1 cm) was place upside down on the soil surface and pressed into the soil. A trowel was pushed below the base and together with the samples it was lifted from the ground and carefully turned around to remove surplus soil. The sample was closed with the upper lid of the perti dish, sealed with parafilm and tape and labelled. All samples were taken in dry state. If wet samples had been taken, these would have needed to be fully dried before sealing. The mass of those samples ranged between 25 and 35 g. The biocrust samples consist of a few mm of biocrust and

the underlying soil (total height about 1 cm). For the chamber measurements the whole samples were used, so that the biocrust was intact/undamaged. The samples were measured in an untreated manner and only the samples for the nutrient and chlorophyll analysis were ground.

The storage (time) has a no significant impact on nutrient content and hence HONO and NO emissions, as co-authors of this study have investigated recently (see also comment to reviewer #1). Great care was taken that the biocrusts were stored in a dry state and in the dark to make sure that they are inactive.

The revised manuscript states:

Chapter 2.1: "Each sample was collected in dry state in a plastic petri dish (diameter 5.5. cm, height 1 cm), sealed and stored in the dark at room temperature until further analysis (storage time less than 15 weeks). Based on previous experiments in our laboratory, it can be anticipated that the sample's chemical (nutrient content) and biological (chlorophyll content) properties were not deteriorated during storage (a manuscript on this study will be submitted soon)."

Chapter 2.4: "Intact soil and biocrust samples (25-35 g in a plastic petri dish with 5.5 cm diameter and about 1 cm height) were wetted with 8-13 g of pure water (18.2 MΩ) up to full water holding capacity and placed into a dynamic Teflon film chamber…Intact (biocrust) samples consist of a few mm of the biocrust and the underlying soil."

In the original manuscript (chapter 2.3) it was already stated, that the samples were ground for nutrient and chlorophyll analysis ("…the samples comprised of soil and its biocrust-cover were gently ground…", "Ground samples were extracted twice…")

**Comment:**

*The sampling procedure and consistency/physical properties of the sample that was placed in the chambers is critical for this type of study. There has been a debate among researchers about how representative gas fluxes are for sieved or cored soil samples of environmental conditions. Previous studies suggest that such laboratory studies of soil cores give similar flux measurements as eddy covariance for grassland soils. In such soils, the surface porosity can be considered to be more similar to porosity of soil just below the surface and arguments could be made that gas exchange from soil in the field and in laboratory cores might be similar. However, biocrusts may present a particularly difficult biome to sample in this way since the intact soil and disturbed soil may have very different structural properties. The physical structure of these surfaces is defined by a network of filamentous growth and biomass that creates a hard crust that is often an impermeable layer that may impact gas exchange. These structural features are known to form hard crusts that prevent soil erosion in sensitive arid ecosystems. The soil exposed when soil is extracted as a core or sieved soil may provide a means to bypass surface structural properties that hinder gas exchange. Do the authors have any evidence to suggest that their method of sampling did not impact gas exchange from these samples? It is important to demonstrate that the results are close to reality and can be used for the type of scaled up estimation performed at the end of the manuscript.*

**Response:**

In order to study the emissions from biocrusts, the samples must be intact as sieving would destroy the crust network/community which probably has an impact on exchange processes. To be as much representative as possible, we made sure that the whole core samples were not sieved or otherwise modified. Although the crust surface, especially with cyanobacteria, is quiet hard, it allows for exchange of nitrogen gases (please see Weber et al., 2015). Earlier studies have shown that NO emissions obtained by the dynamic chamber are consistent with flux measurements in the field (van Dijk et al., 2002; Rummel et al., 2002).

Added to the manuscript in chapter 2.4 (page 4, lines 34-35): "The dynamic chamber method…and in general showed good agreement with flux measurements in the field (van Dijk et al., 2002; Rummel et al., 2002)."

**Comment:**

*While the physical appearance of biological soil crusts is a useful classification tool, it does not provide any information on the actual nitrification processes that occur in or below the biocrust and may be responsible for controlling soil emissions of HONO and NO. Much of the molecular biology that is important for atmosphere-*

*land interactions is likely occurring just belowground (i.e., below the crust that is visible at the surface). It is also misleading to focus solely on the moss, lichen, actinobacteria, which are not the direct sources of these gases. Although biocrusts affect nutrient availability via N fixation, it is their possible associations with ammonia (and nitrite) oxidizing microbes (bacterial and archaea) that ultimately convert the fixed nitrogen to nitrite and nitrate. The current study does not consider the role of ammonia oxidizing microbes in association with biocrusts or the other surface types in the area. These microorganisms are not limited to living within or under biocrusts, but are present in most other soil types to differing degrees. It does make sense that such nitrifying organisms will thrive where their substrates are abundant. However, there are numerous other soil types where this may be the case. Further, there may be many other soil organisms that compete with nitrifiers for their substrates, that may reduce their abundance in soil that would seemingly favor nitrifier populations. The literature that does exist (e.g., Frontiers in Microbiology 2016, doi:10.3389/fmicb.2016.00505) on biocrust-nitrifier associations suggests that biocrusts do not necessarily host a greater abundance of ammonia oxidizing organisms compared to soil supporting trees, nitrogen fixing shrubs, etc. This is an important topic to address.*

**Response:**

Thank you very much for that very good comment. The focus of the current study was to representatively quantify the HONO and NO emissions from the soil/biocrusts and to estimate their importance to the HONO budget by comparing these with the observed missing source. The underlying biological mechanisms were not focus of the current study and thus not discussed at greater detail.

It indeed is true that the dominating photoautotrophic compound doesn't tell us much about the microbial community below these, although, as suggested by the referee, these may have an effect on the belowground microbial community. A problem is, that different biocrust types could be distinguished in the field based on the dominating photoautotrophic compound, whereas microbial communities below the surface could not be determined by non-destructive methods. Within biocrusts, nitrification (and other nitrogen cycling processes) are expected to occur and the relevance of these processes is expected to be also substrate dependent (i.e. depending on the amount of ammonia present for nitrifiers to be used). We agree with the referee that these mechanisms are not restricted to biocrusts, but universally may also occur in non-crusted soils.

In the revised manuscript the following was added:

Chapter 1: "But much of the molecular biology/chemistry that is important for atmosphere-land interactions is likely occurring just below the crust (that is visible at the surface)."

Chapter 3.2: "The different biocrust types were distinguished in the field based on the dominating phototrophic compound but which provides no information about the microbial community below or about the magnitude of (de)nitrification processes. The microbial community couldn´t be determined by non-destructive methods."

Chapter 3.3: "Furthermore it was not possible to determine the microbial community below the biocrust or in bare soil. Although biocrusts increase nutrient availability via N fixation, it is their possible associations with ammonia oxidizing microbes (bacterial and archaea) that finally convert the fixed nitrogen to nitrite and nitrate. Nitrification and other nitrogen cycling processes are not restricted to biocrusts, but can also occur in non-crusted soils. The relevance of these processes is expected to depend on substrate richness (i.e. amount of ammonium available for nitrifiers)."

**Comment:**

*Related to this, Figure 3 of the current manuscript demonstrates that there are other soil types throughout the landscape characterized by stones, litter, and vegetation cover that do not have associated flux values and were not included in the final conclusion regarding relative importance of biocrusts in HONO and NO emissions. The model only considered the approximately 45% of the surface types whose fluxes were characterized. It is possible that fluxes in the other soil types had as high or higher fluxes? If so, would this not make the estimate of contributions of soil emissions to overall atmospheric composition higher and possibly overshoot the Cyprus HONO budget determined in the field campaign? Indeed, Figure 8 is somewhat misleading since it must be noted that F\* only refers to the total HONO and NO flux associated with the 45% of surface types that were actual studied. It is very possible that the pie charts would look very different if other surfaces types were considered. So there is a large uncertainty here.*

**Response:**

To the best of our knowledge, there are no HONO and NO emissions from vascular vegetation, litter and stones. We also thoroughly searched the literature and did not find any publications showing emissions from these surface covers. This was also stated in the original manuscript "To the best of our knowledge, no data on reactive nitrogen emissions from vascular vegetation and plant litter have been published yet." (see original manuscript page 6, lines 14-15). Thus, we are very confident that F* (accounting for 45% of the total surface) represents the effective total emission from ground surface.

**Comment:**

*In my opinion, a satisfying or conclusive connection between the soil emissions of NO and HONO and biocrusts has not been made. The most one can conclude from this study is that volatilization from soil bound nitrite could contribute to the NO and HONO measured in the air above the soil. Indeed, it may have been useful for the authors to include a better discussion of why they can rule out long range transport and atmospheric deposition of nitrate and NOx over time as the source of HONO and NO precursors to this soil. Even though this particular area of Cyprus may have a low population, is possible for it to accumulate anthropogenic inputs from population centers surrounding the Mediterranean basin over time? One is left wondering whether the results support the paper's title and the conclusions it suggests.*

**Response:**

During the CYPHEX campaign (Meusel et al. 2016) very low NOx levels were detected (< 1ppb). Therefore deposition of NOx to the ground which could be converted into $NO_2^-/NO_3^-$ and HONO was excluded as a relevant source. Also nitrate and ammonium concentrations in aerosol particles, ranging from 0.05-0.35 µg m$^{-3}$ and 0.1-4 µg m$^{-3}$, respectively, (measured during CYPHEX) were too low to significantly account for HONO formation. It is not expected that the concentrations in this region are usually higher than found during CYPHEX. So deposition of reactive nitrogen species to the ground is low, and biologic processes (nitrification, denitrification) are the only reasonable explanations for HONO and NO emissions. The emissions were shown to be clearly linked to nutrient and particularly nitrite content Fig. 6), which in the current study seem to be driving HONO and NO emissions of crusted and non-crusted soils.

Please also check our response of referee #3 on [HONO]* calculations to answer the issue about simple volatilization from soil.

In the revised manuscript a short discussion was added:

"Nevertheless, a dominant contribution from microbial activity to the nutrient content is anticipated. Long range transport and atmospheric deposition of NOx and nitrate/nitrite/ammonium can be excluded to be a dominant source of HONO and NO precursors in local soil, as the observed concentrations in Cyprus ambient air were very low (Meusel et al., 2016; Kleanthous et al., 2014)."

**Comment:**

*Lastly, Figure 2 presents a month of meteorological data (air and surface T, air and surface %RH, and precipitation) at the site for the month before samples were taken. The data features prominantly as Figure 2, yet is not used. So, it is unclear why an entire figure was devoted to this data when averages for these values during the time of sampling could have been provided in the text.*

**Response:**

Agree, we now moved this figure to the supplement. Instead, we show a diel pattern of the mean surface temperature and RH with an estimated diel pattern for soil water content and simulated emissions (Fig. R2 or 8 in the revised manuscript; as suggested from referee #1).

**Comment:**

*In conclusion, I feel that the strengths of this manuscript are that it is mostly well written and provides supporting evidence for the fact that soil emissions could have impacted the NOx and HONO budget during the CYPHEX 2014 field campaign. Weaknesses include: (i) there is minimal evidence from this study to support that*

*the emissions are biological in nature (outside of the fact that the flux vs. soil moisture plot matches those of studies on pure cultures of ammonia oxidizing bacteria, Oswald et al.) and (ii) there is less evidence that the actual biocrusts are the dominant HONO and NO sources in this area since we have no data on emissions from 55% of the other surface types present in the study area. Care must be taken here to not draw too much information from these results. The approach described in this paper is not unique; its novelty is related to providing data on soil HONO and NO emissions from understudied region of the globe. Due to its limited scope, this study would have been better suited as supporting data to include in the field campaign paper by Meusel et al. 2016. It may be possible for this study to stand on its own if the above concerns are appropriately addressed in a revised manuscript.*

**Response:**

The aim of this study was not to prove the biological role or to characterize the biological mechanisms of HONO and NO emissions, but to show that soil and biocrust-covered soil in a remote (low pollution) area are an important HONO source (not differentiating between biological or solely physical exchange processes). The residual 55% of the surface coverage which was not studied in detail are very unlikely to emit significant amounts of HONO or NO, and no single study has indicated such an emission, so that the calculated F* is considered to be representative for the whole (local) surface.

**Reference**

Kleanthous, S., Vrekoussis, M., Mihalopoulos, N., Kalabokas, P., and Lelieveld, J.: On the temporal and spatial variation of ozone in Cyprus, Science of The Total Environment, 476–477, 677-687, http://dx.doi.org/10.1016/j.scitotenv.2013.12.101, 2014.

Meusel, H., Kuhn, U., Reiffs, A., Mallik, C., Harder, H., Martinez, M., Schuladen, J., Bohn, B., Parchatka, U., Crowley, J. N., Fischer, H., Tomsche, L., Novelli, A., Hoffmann, T., Janssen, R. H. H., Hartogensis, O., Pikridas, M., Vrekoussis, M., Bourtsoukidis, E., Weber, B., Lelieveld, J., Williams, J., Pöschl, U., Cheng, Y., and Su, H.: Daytime formation of nitrous acid at a coastal remote site in Cyprus indicating a common ground source of atmospheric HONO and NO, Atmos. Chem. Phys., 16, 14475-14493, 10.5194/acp-16-14475-2016, 2016.

Rummel, U., Ammann, C., Gut, A., Meixner, F. X., and Andreae, M. O.: Eddy covariance measurements of nitric oxide flux within an Amazonian rain forest, Journal of Geophysical Research: Atmospheres, 107, LBA 17-11-LBA 17-19, 10.1029/2001JD000520, 2002.

van Dijk, S. M., Gut, A., Kirkman, G. A., Gomes, B. M., Meixner, F. X., and Andreae, M. O.: Biogenic NO emissions from forest and pasture soils: Relating laboratory studies to field measurements, Journal of Geophysical Research: Atmospheres, 107, LBA 25-21-LBA 25-11, 10.1029/2001JD000358, 2002.

Weber, B., Wu, D., Tamm, A., Ruckteschler, N., Rodriguez-Caballero, E., Steinkamp, J., Meusel, H., Elbert, W., Behrendt, T., Soergel, M., Cheng, Y., Crutzen, P. J., Su, H., and Poeschl, U.: Biological soil crusts accelerate the nitrogen cycle through large NO and HONO emissions in drylands, Proceedings of the National Academy of Sciences of the United States of America, 112, 15384-15389, 10.1073/pnas.1515818112, 2015.

---

## Author Comment (AC3) · 9 Oct 2017

**Summary:**
*Soil samples used in this work are from soils collected from the field, manipulated in a controlled lab environment, and then measured fluxes extrapolated to compare with the missing HONO source calculated for the CYPHEX field campaign in the same location. Soils were collected and categorized from a gridded sampling scheme. HONO and NO fluxes were measured from the soils in replicates in order to quantify which surface soil community members, if any, were responsible for the majority of the HONO fluxes observed. The authors performed nice controlled experiments in the lab and found some interesting conclusions, counter to previous findings in similar soils by this group. The manuscript may be acceptable for publication in Atmospheric Chemistry and Physics, subject to a number of concerns being addressed.*

**Comment:**
*There is no experimental control of soils devoid of microbial activity for each soil type. One would think it necessary to fumigate (or otherwise kill) soil samples of each type to control for biotic versus abiotic HONO and NO emissions, yet this is not presented. If possible, the authors should consider acquiring this data and adding it to the manuscript for comparison and correction of the dataset.*

**Response:**
The main purpose of this study is to investigate if Cyprus soil and representative biocrust covers are indeed an important source for HONO, as was assumed from atmospheric observations in an earlier paper by Meusel et al. (2016). The role of biological activities versus physical emission was not focus of the present study, but was proven in earlier studies. As shown by Oswald et al. (2013), natural soils emit much more HONO and NO than sterilized samples, and also Weber et al. (2015) found strongly decreased emission upon sterilization, pointing to a biological emission process.
We add the following note into the introduction of the manuscript: "It was found that sterilized soil emit lower amounts of reactive nitrogen than natural soil (Oswald et al., 2013; Weber et al., 2015)."

**Comment:**
*The consideration of the effects of the measured soil pH on HONO release using the method of the Su et al. (2011) work is not considered in the interpretation of the data. What proportion of the emissions measured in each case can be ascribed to simple partitioning? What effect does this have on comparisons between soil types considered in this work when abiotic exchange is estimated versus measured (see comment above) in the experimentally measured fluxes? A major concern is that if abiotic partitioning from dead soils is not favored by calculation from bulk pH measurements and nutrient loadings (i.e. soil pH » pKa HONO) then another emission mechanism is active and should be considered/discussed.*

**Response:**
Following the suggestion by the referee, [HONO]* was calculated according to Su et al. (2011) for 2 different $NO_2^-$ contents in the given range of observed $NO_2^-$ content and their observed mean pH-value. In this calculation the absolute amount of nitrite was assumed to be constant, i.e., with lower soil water content the liquid phase nitrite concentration increases (please note that we did not find significant differences between nitrite concentrations before and after the chamber trace gas exchange experiments).

As expected, the calculated equilibrium concentration [HONO]* shows a positive dependence on nitrite concentration (compare dashed lines in Fig. R3) and pH (compare dashed red and orange lines in Fig. R3). At the optimum soil water content of 10-20 % (mass $H_2O$/mass soil) or 25-35% whc, respectively, the calculated [HONO]* (at pH = 7) is only about 5-10% of the one observed by chamber measurements. For slightly lower pH the calculated [HONO]* increase, and contribute about 17% to the measured.

[Figure]

Fig. R4 (S3): Calculated [HONO]* for two different $NO_2^-$ concentrations at pH 7 and pH 6.5 (dashed lines) in comparison with measured [HONO]* for two samples with similar $NO_2^-$ content (solid lines) vs the gravimetric soil water content.

Indeed, as stated in Su et al. (2011) the calculated [HONO]* based on nitrite partitioning may deviate from the measured values due to the non-ideal solution behavior (adsorption, Kelvin and solute interaction effects on gas/liquid partitioning). Thus the agreement between simple models (based on ideal solution assumption) and measurements cannot be used to discriminate the physical and biological processes. We also want to clarify that the soil emission proposed by Su et al. (2011) is not an abiotic processes. Their conclusion is that biogenic nitrite in soil can be emitted to the atmosphere, of which the transport or partitioning is also subject to other physio-chemical processes like other nitrogen containing gases (e.g., NO, see fig. R4).

[Figure]

Fig. R5 (Fig 1 of Su et al., 2011): Coupling of atmospheric HONO with soil nitrite. Red arrows represent the multiphase processes linking gaseous HONO and soil nitrite (acid-base reaction and phase partitioning), green arrows represent biological processes, orange arrows represent heterogeneous chemical reactions converting $NO_2$ and $HNO_3$ into HONO, and blue arrows represent other related physicochemical processes in the N cycle.

This consideration is now also added to the revised version of the manuscript (end of chapter 3.3):
"Since most of the samples were slightly alkaline and only moss samples were slightly acidic, no effect of pH could be observed. But in general it is expected that with higher nutrient and lower pH values HONO emission is increased by simple partitioning processes (Su et al., 2011). The simulated equilibrium concentration at soil surface [HONO]* (equation see Su et al., 2011) is much lower than the measured one (see supplement Fig. S3). This deviation is probably based on the non-ideal behavior of the soil samples (adsorption, Kelvin and solute interaction effects on gas/liquid partitioning). But this method does not allow differentiation between physical or biological nitrite production processes."

**Comment:**
*Scaling to atmospheric relevance for the field campaign is very interesting, but inappropriate to include in the title of the manuscript. The linkage between microbial activity and HONO and NO emissions is of growing importance to constrain and these lab-based measurements help to do so. However, the uncertainty in the*

*extrapolation of the data to the field campaign observations of the missing daytime HONO source covers an order of magnitude range, which means at the low end of the estimate these*
*processes account for < 10 % of the daytime HONO source. The authors do not discuss this limitation and should do so. The title of the manuscript should also remove the HONO budget closure implications due to this significant uncertainty. An important consideration here is one that has been reported and discussed much in the atmospheric community over the past 5 years and that is the vertical structure of HONO, and by proxy, the daytime HONO source strength near the ground surface. Using soil HONO emissions to scale to measurements made nearly 6 m above the ground may add additional error in the budget closure calculation as the perceived missing source changes with height. This topic and the relevant references would be a worthwhile addition to this component of the discussion as there are several reports on vertical structure of HONO in arid agricultural and rural regions in the US.*

**Response:**
The title doesn´t really imply a HONO budget closure, it only indicates that soil emission can be a dominant source. Nevertheless, following the reviewer's suggestion, we now tuned the title reading "… represents an important source…".
Indeed, the range of HONO production given in the original manuscript had a high uncertainty, spanning an order of magnitude (based on the lowermost and uppermost fluxes observed in the lab). According to the suggestion of referee #1, we now give a more detailed best estimate, including more site-specific input variables (diel trend of temperature and soil water content). This confines the estimated source strength to $6 \times 10^4$-$1.1 \times 10^6$ $cm^{-3}\,s^{-1}$.
Anyhow, in agreement to the referee's objection we now better emphasize the uncertainty of the newly calculated HONO emissions in the discussion section of the revised version of the manuscript, like "… the emissions at 25°C and a swc of 10% whc would span a wide range between $1.1 \times 10^5$ and $9.6 \times 10^5$ $cm^{-3}\,s^{-1}$, covering 9 to 73% of the missing mean source of $1.3 \times 10^6$ $cm^{-3}\,s^{-1}$ observed in the field (Meusel et al., 2016)…" (see also the response of referee #1).

It is true that there is a gradient in HONO concentration in the atmosphere with higher concentration near the ground. Our estimate does not include a respective chemistry-transport model (accounting for vertical gradients of atmospheric sinks and sources), nor accounts for the existence of a vertical profile of concentrations. The ground-based source was calculated for a boundary layer height of 300 m above ground level, found typical for Cyprus during the campaign 2014 (for details please see Meusel et al., 2016). The method is according to, e.g., Stemmler et al. (2006), who used homogeneous mixed air columns between 150 and 430 m for their calculations of a surface-based HONO source. A recent study calculated the height over a rural basin in Utah, USA, at which the influence of HONO surface fluxes on the total HONO column becomes negligible. At a height of 273 m the impact of the surface flux to the HONO budget was less than 1 ppt (Tsai et al., 2017).
We now discuss it in the manuscript:
"Field observations (VandenBoer et al., 2013; Zhang et al., 2009; Tsai et al. 2017) as well as model results (Wong et al., 2013) showed that HONO concentrations typically decrease exponentially from the surface upwards. Eq. 7 does not include a chemistry-transport model, nor accounts for the existence of a vertical profile of concentrations, which may bias the calculation on HONO source strength. But the method for predicting the ground source using homogeneous mixed air columns is consistent with other recent studies (Stemmler et al., 2006; Tsai et al., 2017). Tsai et al. (2017) clearly showed the presence of an important ground source of daytime HONO at a rural basin in Utah, during wintertime (no snow, low temperatures). They inferred that ground surface fluxes may account for 63±32% of the unidentified HONO daytime source throughout the day. HONO fluxes of up to 7.4 ng $m^{-2}\,s^{-1}$ (Fig. 8, lower panel) determined in this study are comparable to HONO fluxes found in other regions, e.g., 2.7 ng $m^{-2}\,s^{-1}$ reported for the northern Michigan forest canopy (Zhang et al., 2009; Zhou et al., 2011), the average daytime HONO flux of 7.0 ng $m^{-2}\,s^{-1}$ measured over an agricultural field in Bakersfield (Ren et al., 2011), and the average HONO flux of about 11.6 ng $m^{-2}\,s^{-1}$) measured by Tsai et al. (2017). In contrast to the present study, the latter concluded that, under the prevailing high NOx conditions, the respective HONO formation was related to solar radiation and $NO_2$ mixing ratios, such as photo-enhanced conversion of $NO_2$ or nitrate photolysis on the ground. This can be ruled out in this study, as pure air (no $NO_2$) was used to purge the chambers and no light was applied. "

**Comment:**

*The quality of the nutrient data cannot be determined as no accuracy, precision, or detection limit values are presented. The data table in the supplement reports measurements of zero, which should be represented by '< LOD' and the detection limits determined by the experimental runs calculated (do not use the instrument manufacturer's stated values). My concern is that some of the measurements made on the samples are near the detection limits and therefore highly uncertain, which may confound the comparisons made throughout the manuscript. The authors should also be more cautious in their reported values from these measurements. Are these analytically certain to so many significant digits?*

**Response:**

Agreed, the reported data in the supplement table on nutrient levels close to 0 are now changed to "<LOD". The detection limits were checked again and were 0.012, 0.051, and 0.015 mg L$^{-1}$ for $NO_2^-$, $NO_3^-$ and $NH_4^+$, respectively. Transforming the unit to mg kg$^{-1}$ results in detection limits of 0.014, 0.046 and 0.047 mg kg$^{-1}$ for $NO_2^-$-N, $NO_3^-$-N, and $NH_4^+$-N. Most of the nutrient levels are well above these levels, no concerns from your side.

The detection limits are now added into the revised version of the manuscript: "The detection limits were 0.014, 0.046 and 0.047 mg kg-1 for $NO_2^-$-N, $NO_3^-$-N and $NH_4^+$-N."

**Comment:**

*The experiments performed on nutrient content before and after experiments (Figure 4) is good to have performed, but this data does not need to be presented in the figure and would help make this a cleaner plot. The data can be replaced with one sentence in the manuscript stating that the nutrient levels were not different in the soils by performing the flux experiments. To that end, this suggests that the full set of measurements of these nutrients could be pooled and improve the statistical analyses performed, as it would reduce the standard error in the measurements. Further to this point, where replicate measurements have been made, the authors should be presenting the standard error of the mean and not the standard deviation as it aids in connecting the reader to the statistical results. The results and discussion surrounding the purpose, method selection, and outcomes of these statistical tests needs to be improved either through clarity in existing sections or expansion of the text.*

**Response:**

Thanks for pointing out. The figures are modified accordingly. Figure 4a now shows the mean value of nutrient content of all samples and are not separated into samples without and after flux measurements. Error bars (in Fig 4a, b, c; 6) now indicate the standard error of the mean and not the standard deviation. Also in the text the standard deviation was changed to the standard error.

[Figure]

*New Fig 4 (revised manuscript, fig.3)*: a: nutrient content *for all samples*, star indicate outlier in $NH_4^+$ content (for light BSC), for a-c error bar indicate *standard error of the mean*, letters indicate significant difference (p=0.05, of log data).

[Figure]

*New Fig 6 (revised manuscript, fig. 5): …error bars indicate standard error of the mean…*

**Comment:**
*Details on linear regression in Figure 7 needs to be presented. There is presumably large uncertainty in both measurements being compared and the appropriate considerations must be included prior to assessing the relationship between them, along with the associated uncertainty in the result. Building on this, the direction of*

*the trend and also its associated uncertainty from the regression fit may be more telling towards whether the strength of the coefficient is truly robust or being limited by the sample size available.*

**Response:**
Regressions considering x-, and y- errors were performed as suggested by the referee. These were done by using the excel sheet for bivariate regressions provided by Cantrell, 2008. The errors were related to the uncertainties of the measurements (10%). In this method smaller values have smaller errors and therefore are weighted more. For nutrient levels < detection limit the value was set to the detection limit.
The coefficient of determination ($R^2$) decreased slightly when errors are considered. But the correlation is still significant (see table below; error bars in the figures indicate the 10% uncertainty).

Table 1: Results of regressions performed by excel (Cantrell, 2008):

| **Nitrite – HONO** (< LOD = LOD) | **Nitrite-NO** (< LOD = LOD) |
|---|---|
| a) Errors (x , y) 10% of measured value | a) Errors (x , y) 10% of measured value |
| Regression:  y = 410.9*x+7.4 ($R^2$ = 0.867)  standard error m = 0.078, b = 6.52 | Regression:  y = 216.1*x+11.5 ($R^2$ = 0.76)  standard error m = 0.017, b = 5.04 |
| b) Without errors | b) Without errors |
| Regression:  y=394.14*x+0.804 ($R^2$=0.885)  standard error m =30.4, b=5.5 | Regression:  y=213.88.14*x+6.94 ($R^2$=0.776)  standard error m =24.5, b=4.4 |

[Figure]

| **Nitrate – HONO** (< LOD = LOD) | **Nitrate-NO** (< LOD = LOD) |
|---|---|
| a) Errors (x , y) 10% of measured value | a) Errors (x , y) 10% of measured value |
| Regression:  y = 25.97*x+27.46($R^2$ = 0.484)  standard error m = 0.060, b = 11.3 | Regression:  y = 23.33*x+5.96 ($R^2$ = 0.475)  standard error m = 0.074, b = 9.16 |
| b) Without errors | b) Without errors |
| Regression:  y=22.68*x+10.74 ($R^2$=0.674)  standard error m =3.46, b=6.19 | Regression:  y=17.19*x+12.55 ($R^2$=0.547)  standard error m =3.31, b=6.02 |

[Figure]

As the modified regressions are very similar to the original ones, we decided to keep the simple regressions, but added the regression details (lines and formula) into the plots of the revised manuscript.

[Figure]

(new Fig. 7, in the revised manuscript fig. 6)

**Comment:**
*Figure 2 needs to be streamlined to present the relevant information for the contents of the manuscript. The level of detail here is not necessary.*

**Response:**
Figure 2 was moved to the supplement and a diel pattern of soil or surface climate is shown now in Fig 8.

**Minor Comments:**
*Fix tense and plurality issues throughout the manuscript.*
*Many sentences have issues with comma splicing, making them long and the purpose of the sentence difficult to follow.*

**Response:**
We read through the manuscript again and carefully checked the tense and rewrote several sentences.

**Reference**
Cantrell, C. A.: Technical Note: Review of methods for linear least-squares fitting of data and application to atmospheric chemistry problems, Atmos. Chem. Phys., 8, 5477-5487, 10.5194/acp-8-5477-2008, 2008.
Meusel, H., Kuhn, U., Reiffs, A., Mallik, C., Harder, H., Martinez, M., Schuladen, J., Bohn, B., Parchatka, U., Crowley, J. N., Fischer, H., Tomsche, L., Novelli, A., Hoffmann, T., Janssen, R. H. H., Hartogensis, O., Pikridas, M., Vrekoussis, M., Bourtsoukidis, E., Weber, B., Lelieveld, J., Williams, J., Pöschl, U., Cheng, Y., and Su, H.: Daytime formation of nitrous acid at a coastal remote site in Cyprus indicating a common ground source of atmospheric HONO and NO, Atmos. Chem. Phys., 16, 14475-14493, 10.5194/acp-16-14475-2016, 2016.
Oswald, R., Behrendt, T., Ermel, M., Wu, D., Su, H., Cheng, Y., Breuninger, C., Moravek, A., Mougin, E., Delon, C., Loubet, B., Pommerening-Roeser, A., Soergel, M., Poeschl, U., Hoffmann, T., Andreae, M. O., Meixner, F. X., and Trebs, I.: HONO Emissions from Soil Bacteria as a Major Source of Atmospheric Reactive Nitrogen, Science, 341, 1233-1235, 10.1126/science.1242266, 2013.
Ren, X., Sanders, J. E., Rajendran, A., Weber, R. J., Goldstein, A. H., Pusede, S. E., Browne, E. C., Min, K. E., and Cohen, R. C.: A relaxed eddy accumulation system for measuring vertical fluxes of nitrous acid, Atmospheric Measurement Techniques, 4, 2093-2103, 10.5194/amt-4-2093-2011, 2011.
Stemmler, K., Ammann, M., Donders, C., Kleffmann, J., and George, C.: Photosensitized reduction of nitrogen dioxide on humic acid as a source of nitrous acid, Nature, 440, 195-198, 10.1038/nature04603, 2006.
Su, H., Cheng, Y., Oswald, R., Behrendt, T., Trebs, I., Meixner, F. X., Andreae, M. O., Cheng, P., Zhang, Y., and Poeschl, U.: Soil Nitrite as a Source of Atmospheric HONO and OH Radicals, Science, 333, 1616-1618, 10.1126/science.1207687, 2011.
Tsai, C., Spolaor, M., Colosimo, S. F., Pikelnaya, O., Cheung, R., Williams, E., Gilman, J. B., Lerner, B. M., Zamora, R. J., Warneke, C., Roberts, J. M., Ahmadov, R., de Gouw, J., Bates, T., Quinn, P. K., and Stutz, J.: Nitrous acid formation in a snow-free wintertime polluted rural area, Atmos. Chem. Phys. Discuss., 2017, 1-37, 10.5194/acp-2017-648, 2017.
VandenBoer, T. C., Brown, S. S., Murphy, J. G., Keene, W. C., Young, C. J., Pszenny, A. A. P., Kim, S., Warneke, C., de Gouw, J. A., Maben, J. R., Wagner, N. L., Riedel, T. P., Thornton, J. A., Wolfe, D. E., Dubé, W. P., Öztürk, F., Brock, C. A., Grossberg, N., Lefer, B., Lerner, B., Middlebrook, A. M., and Roberts, J. M.: Understanding the role of the ground surface in HONO vertical structure: High resolution vertical profiles during NACHTT-11, Journal of Geophysical Research: Atmospheres, 118, 10,155-110,171, 10.1002/jgrd.50721, 2013.
Weber, B., Wu, D., Tamm, A., Ruckteschler, N., Rodriguez-Caballero, E., Steinkamp, J., Meusel, H., Elbert, W., Behrendt, T., Soergel, M., Cheng, Y., Crutzen, P. J., Su, H., and Poeschi, U.: Biological soil crusts accelerate the nitrogen cycle through large NO and HONO emissions in drylands, Proceedings of the National Academy of Sciences of the United States of America, 112, 15384-15389, 10.1073/pnas.1515818112, 2015.
Wong, K. W., Tsai, C., Lefer, B., Grossberg, N., and Stutz, J.: Modeling of daytime HONO vertical gradients during SHARP 2009, Atmospheric Chemistry and Physics, 13, 3587-3601, 10.5194/acp-13-3587-2013, 2013.
Zhang, N., Zhou, X. L., Shepson, P. B., Gao, H. L., Alaghmand, M., and Stirm, B.: Aircraft measurement of HONO vertical profiles over a forested region, Geophysical Research Letters, 36, 10.1029/2009gl038999, 2009.
Zhou, X., Zhang, N., TerAvest, M., Tang, D., Hou, J., Bertman, S., Alaghmand, M., Shepson, P. B., Carroll, M. A., Griffith, S., Dusanter, S., and Stevens, P. S.: Nitric acid photolysis on forest canopy surface as a source for tropospheric nitrous acid, Nature Geoscience, 4, 440-443, 10.1038/ngeo1164, 2011.

---

## Author Response (AR2)

**Referee (#2) comment on the revised manuscript:**

**Emission of nitrous acid from soil and biological soil crusts represents an important source of HONO in the remote atmosphere in Cyprus**

By Hannah Meusel et al.

**General comment:**

*The manuscript has been revised. I feel the link between these laboratory studies and the published field campaign is still surrounded in uncertainty; there remain questions regarding the identity of the missing HONO source inferred from field data. For example, the field data published by Meusel et al. 2016 showed correlations between the unknown HONO source strength and J(NO2)\*NO2 (R2 = 0.813) in addition to correlations between daytime HONO and NO (r2 = 0.60), but they concluded that the HONO source was independent of NO2 and rather related to (biological) emissions from soil. Correlations between HONO and J(NO2)\*NO2 have in the past been used to argue in favor of a photochemical NO2 source in higher NOx regimes (e.g., Laufs 2017). I am still not convinced by a regression analysis that long-term accumulation of N nutrients via atmospheric deposition to the high surface area ground surface can be neglected as a source of soil HONO fluxes inferred during the Cyprus field study. In arid areas, lack of rain that would leach ions to ground water means that N can accumulate over long periods of time to high levels in the upper levels of the soil column. Regardless of the imperfections in extrapolating the data to the field campaign results, the laboratory studies show it is possible that soil emissions (whether biological or from atmospheric deposition) are potential contributors to HONO measured during the field campaign. Data related to HONO soil emissions is still sparse, so the data presented here will be useful to the atmospheric community. For this reason, I think it could be published in ACP after the following points have been addressed.*

**Response:**

Though the missing HONO source was well correlated with $J(NO_2)*NO_2$, a more pronounced HONO source independent from $NO_2$ was proposed for Cyprus as respective atmospheric $NO_2$ concentrations were too low to significantly contribute to the missing source. Also the deposition of reactive nitrogen species ($NO_3^-$, $NO_2^-$, NO, $NO_2$) was excluded due to low atmospheric concentrations in this area and time. But we cannot exclude long term accumulation of those as the referee proposed. Nevertheless is emission from soil an important HONO source on Cyprus.

We modified the manuscript (chapter 3.3) as follows: "…Long range transport and instantaneous atmospheric deposition of NOx and nitrate/nitrite/ammonium can be excluded to be a dominant source of HONO and NO precursors in local soil, as the observed concentrations in Cyprus ambient air were very low (Meusel et al., 2016; Kleanthous et al., 2014). A dominant contribution from microbial activity to the nutrient content is anticipated, although long-term atmospheric accumulation of nutrients in the soil prior to the field campaign cannot be excluded."

**Comment 1:**

*The lab-based HONO and NO fluxes are strongly dependent on soil moisture with maximum emission rates between 17-33% and 20-36% WHC for HONO and NO, respectively. Do the authors have any information on water content of the soil during the field campaign to say where they are on the flux-water content curve in the field? A quantitative discussion of how the soil moisture regime of the lab-based study compares to actual field conditions would greatly enhance the argument that soil emissions were important during the field campaign.*

**Response:**

Unfortunately we didn´t measure the soil water content during the first field campaign (CYPHEX 2014). But as already described in the manuscript we estimated a quiet dry soil (below 14% WHC) as no rain was detected during the campaign, which is comparable to lab studies in which a soil water content of about 10% WHC was observed at high relative humidity. Respective references are already stated in the text of the manuscript (Likos, 2008; Leelamanie, 2010).

**Comment 2:**

*p. 3, line 30: The authors include the following statement: "Based on previous experiments in our laboratory, it can be anticipated that the sample's chemical (nutrient content) and biological (Chlorophyll content) properties were not deteriorated during storage (a manuscript on this study will be submitted soon)." I have reservations about including such statements in papers. Please provide quantitative data in this manuscript to support this or include a submitted manuscript as review material for evaluation. I feel this could be done without compromising the other manuscript. For example, submitting for the record results from a nutrient/chlorophyll analysis at time of soil sampling in field vs. nutrient levels at time of flux measurement. While I have every reason to trust the authors' statements, I feel that promising a manuscript on this topic should is not acceptable support for the assumption that the sample did not deteriorate over time.*

**Response:**

We follow the referee´s suggestion and removed the statement on the near-future submission on the related manuscript. But unfortunately it is not possible for us to show the results of this study on the impact of storage time and conditions on sample properties (nutrient, chlorophyll content). Instead we stress the fact that this is the most widely used method and storage times are often up to 6 month (chapter 2.1): "… Storage of biocrust samples under dry and dark conditions at room temperature is the most widely spread method, and has been used in many other studies on N-cycling in which samples have been stored even up to 6 month before measurements were performed (Abed et al., 2013; Strauss et al., 2012:, Johnson et al., 2007; Brankatschk et al., 2013)."

**Comment 3:**

*p. 4, line 1: I note that intact soil and biocrust samples in petri dishes were added to the flux chambers. The act of sampling this soil is exposing once covered and intact soil from the underlayer. Considering that diffusion of gases in and out of the surface crust would be lower than broken and exposed soil below, I wonder what proportion of the observed emissions stem from the diffusion from underlayers of exposed soil, and how representative the fluxes are. A previous comparison between eddy covariance fluxes vs. laboratory flux chamber measurements is cited as showing that the methods are comparable, but the soils from that experiment are from the Amazon rain forest, so they are not an ideal comparison. Porosity and composition for those soils will be very different than for the desert soils studied here.*

**Response:**

Microbial activity responsible for production and consumption of reactive nitrogen species in soils (nitrification, denitrification) has been shown to be confined to the uppermost soil layer (e.g., Rudolph et al., 1996). As written in the manuscript, also the biocrusts grow within the uppermost millimeters to centimeters of soil in arid and semi-arid ecosystems. We agree with the referee that the soil diffusivity could have been altered due to our sample treatment. Under dry and hot conditions, like on Cyprus, large macropores and soil cracks can be assumed to have been developed, where relative gas diffusion coefficients can be one order of magnitude higher than in regions without macropores (Allaire et al., 2008; Moldrup et al., 2000). The sample treatment might have increased the number and size of cracks, but they were also present in the natural soil cover during the observed dry conditions prevailing during the CYPHEX campaign. Due to the high variability of diffusion within the uppermost layer of natural soils even in the very small scale, it is difficult to estimate the effect of our soil sample treatments.

In soil in the Amazon Rainforest habitat (cited studies of van Dijk et al., 2002 and Rummel et al., 2002) cracks are normally less frequent, so that the impact of introducing artificial soil diffusion cracks (and exposing soil sublayers) during sampling and sample treatment should have been even stronger as compared to our study. But still the results of flux measurements in the chamber and the non-invasive micrometeorological method in the field for amazon soil were comparable. Therefore we assume that our measured flux or corrected flux F* is representative for this region.

We added following note in the manuscript (chapter 2.4): "…and in general showed good agreement with flux measurements in the field (van Dijk et al., 2002; Rummel et al., 2002). Under the prevailing dry and hot conditions in Cyprus macropores and cracks are likely to be present in the soil layer. It is assumed that during the sampling and sample treatment the number and sizes of soil cracks was not significantly increased so that the diffusivity of gases in the soil samples is comparable to the one in soil in the natural environment."

**Comment 4:**

*p. 8, line 24: Since informative analyses could have be carried out post flux experiment, consider to revise sentence from "…it was not possible to determine the microbial community below the biocrust or in bare soil," to "Determination of the microbial community below the biocrust or in bare soil was not carried out as it was outside the scope of this study."*

**Response:**

Accordingly to the referee´s suggestion we revised the sentence.

**Comment 5:**

*p. 10, line 15-32, Figure 8 labels (and perhaps elsewhere in paper): When reporting number density please include molecule in the units. For example cm-3 s-1 should be written as molecules cm-3 s-1. The omission of 'molecule' has led to confusion among readers in the past, especially among students or readers from other fields.*

**Response:**

Yes, that is right, it could lead to confusion. We changed the unit accordingly.

[revised manuscript text omitted]